# Unified Convergence Theory of Stochastic and Variance-Reduced Cubic Newton Methods

**El Mahdi Chayti**                                                    *el-mahdi.chayti@epfl.ch*
*Machine Learning and Optimization Laboratory (MLO), EPFL*

**Martin Jaggi**                                                       *martin.jaggi@epfl.ch*
*Machine Learning and Optimization Laboratory (MLO), EPFL*

**Nikita Doikov**                                                      *nikita.doikov@epfl.ch*
*Machine Learning and Optimization Laboratory (MLO), EPFL*

**Reviewed on OpenReview:** *https://openreview.net/forum?id=FCs5czlDTr*

## Abstract

We study stochastic Cubic Newton methods for solving general, possibly non-convex minimization problems. We propose a new framework, the *helper framework*, that provides a unified view of the stochastic and variance-reduced second-order algorithms equipped with global complexity guarantees; it can also be applied to learning with auxiliary information. Our helper framework offers the algorithm designer high flexibility for constructing and analyzing stochastic Cubic Newton methods, allowing arbitrary size batches and using noisy and possibly biased estimates of the gradients and Hessians, incorporating both the variance reduction and the lazy Hessian updates. We recover the best-known complexities for the stochastic and variance-reduced Cubic Newton under weak assumptions on the noise. A direct consequence of our theory is the new lazy stochastic second-order method, which significantly improves the arithmetic complexity for large dimension problems. We also establish complexity bounds for the classes of gradient-dominated objectives, including convex and strongly convex problems. For Auxiliary Learning, we show that using a helper (auxiliary function) can outperform training alone if a given similarity measure is small.

## 1 Introduction

In many fields of machine learning, it is common to optimize a function $f(\boldsymbol{x})$ that can be expressed as a finite sum:

$$\min_{\boldsymbol{x} \in \mathbb{R}^d} \Big\{ \ f(\boldsymbol{x}) \ = \ \tfrac{1}{n} \sum_{i=1}^{n} f_i(\boldsymbol{x}) \ \Big\}, \tag{1}$$

or, more generally, as an expectation over some given probability distribution: $f(\boldsymbol{x}) = \mathbb{E}_\zeta \big[ f(\boldsymbol{x}, \zeta) \big]$. When $f$ is non-convex, this problem is especially difficult since finding a global minimum is NP-hard in general (Hillar & Lim, 2013). Hence, the reasonable goal is to look for approximate solutions. The most prominent family of algorithms for solving large-scale problems of the form (1) are the *first-order methods*, such as the Stochastic Gradient Descent (SGD) (Robbins & Monro, 1951; Kiefer & Wolfowitz, 1952). They employ only stochastic gradient information about the objective $f(\boldsymbol{x})$ and guarantee the convergence to a stationary point, which is a point with a small gradient norm.

Nevertheless, when the objective function is non-convex, a stationary point may be a saddle point or even a local maximum, which might not be desirable. Another common issue is that first-order methods typically have a slow convergence rate, especially when the problem is *ill-conditioned*. Therefore, they may not be suitable when high precision for the solution is required.

To address these challenges, we can take into account *second-order information* (the Hessian matrix) and apply Newton's method (see, e.g., (Nesterov, 2018)). Among the many versions of this algorithm, the Cubic Newton method (Nesterov & Polyak, 2006) is one of the most theoretically established. With the Cubic Newton method, we can guarantee *global convergence* to an approximate *second-order* stationary point (in contrast, the pure Newton method without regularization can even diverge when it starts far from a neighborhood of the solution). For a comprehensive historical overview of the different variants of Newton's method, see Polyak (2007). Additionally, the convergence rate of the Cubic Newton is *provably better* than those for the first-order methods.

Recent developments in the theory of second-order optimization methods include the framework of gradient regularization (Mishchenko, 2023; Doikov & Nesterov, 2023), adaptive, universal and super-universal methods (Cartis et al., 2011a;b; Grapiglia & Nesterov, 2017; Doikov et al., 2024), the accelerated Newton methods (Nesterov, 2018), modern analysis of the damped Newton iterations (Hanzely et al., 2022; Hanzely, 2023), and special second-order schemes for self-concordant and quasi-self-concordant functions (Dvurechensky & Nesterov, 2018; Karimireddy et al., 2018; Doikov, 2023). Despite some of the advantageous properties, most of these developments assume that the objective is convex, allowing the use of different regularization techniques and justifying improved convergence rates. In contrast, the cubic regularization technique is applicable to a wide range of problem classes, including non-convex optimization.

Therefore, the theoretical guarantees of the Cubic Newton method seem very appealing for practical applications. However, the basic version of the Cubic Newton requires the exact gradient and Hessian information in each step, which can be very expensive to compute in a large-scale setting; to overcome this issue, several techniques have been proposed:

- One popular approach is to use inexact *stochastic gradient and Hessian estimates* with subsampling (Xu et al., 2016; Kohler & Lucchi, 2017; Xu et al., 2017; Nilesh et al., 2018; Ghadimi et al., 2017; Cartis & Scheinberg, 2018; Agafonov et al., 2020). This technique avoids using the full oracle information, but it typically has a slower convergence rate compared to the exact Cubic Newton.
- *Variance reduction* techniques (Zhou et al., 2019; Wang et al., 2019) combine the advantages of stochastic and exact methods, achieving an improved rate by recomputing the full gradient and Hessian information at some iterations.
- *Lazy Hessian* updates (Shamanskii, 1967; Doikov et al., 2022) utilize a simple idea of reusing an old Hessian for several iterations of a second-order scheme. Indeed, since the cost of computing one Hessian is usually much more expensive than one gradient, it can improve the arithmetic complexity of our methods.
- In addition, exploiting the special structure of the function $f$ (if known) can also be helpful. For instance, *gradient dominated objectives* (Nesterov & Polyak, 2006), a subclass of non-convex functions that have improved convergence rates and can even be shown to converge to the global minimum. Examples of such objectives include convex and star-convex functions, uniformly convex functions, and functions satisfying the PL condition (Polyak, 1963) as a special case. Stochastic algorithms with variance reduction for gradient-dominated functions were studied previously for the first-order methods (see, e.g., Fatkhullin et al. (2022)) and for the second-order methods with cubic regularization in Masiha et al. (2022).

In this work, we revise the current state-of-the-art convergence theory for the stochastic Cubic Newton method and propose a unified and improved complexity guarantees for different versions of the method, which combine the advanced techniques listed above.

Our developments are based on the new *helper framework* for second-order optimization that we present in Section 3. For first-order optimization, a similar in-spirit technique called *learning with auxiliary information* was developed recently in (Chayti & Karimireddy, 2022; Woodworth et al., 2023). Thus, our results can also be seen as a generalization of the Auxiliary Learning paradigm to second-order optimization. However, note that in our second-order case, we have more freedom for choosing the "helper functions" (namely, we use one for the gradients and one for the Hessians). That brings more flexibility into our methods, and it allows, for example, to use the lazy Hessian updates.

Our new helper framework provides us with a unified view of the stochastic and variance-reduced methods and can be used by an algorithm designed to construct new methods. Thus, we show how to recover already known versions of the stochastic Cubic Newton with some of the best convergence rates, as well as present the new *Lazy Stochastic Second-Order Method*, which significantly improves the total arithmetic complexity for large-dimension problems.

**Contributions.**

- We introduce the *helper framework*, which we argue encompasses multiple methods in a unified way. Such methods include stochastic methods, variance reduction, Lazy methods, core sets, and semi-supervised learning.
- This framework covers previous versions of the variance-reduced stochastic Cubic Newton methods with known rates. Moreover, it provides us with new algorithms that employ *Lazy Hessian* updates and significantly improves the arithmetic complexity (for high dimensions), by using the same Hessian snapshot for several steps of the method.
- In the case of Auxiliary learning, we provably show a benefit from using auxiliary tasks as helpers in our framework. In particular, we can replace the smoothness constant with a similarity constant, which might be smaller.
- Moreover, our analysis works both for the general class of non-convex functions, as well as for the classes of gradient-dominated problems, which include *convex* and *uniformly convex* functions. Hence, in particular, we justify new improved complexity bounds for the deterministic Cubic Newton method with Lazy Hessian updates (Doikov et al. (2022)); as well as for the stochastic Cubic Newton algorithms with variance reduction that take into account the total arithmetic cost of the operations (see Table 1).

In the following table, we provide a comparison of the total complexities (in terms of the number of stoch. gradient calls) for finding a point with small gradient norm $\mathbb{E}\|\nabla f(\bar{x})\| \leq \varepsilon$ (non-convex case), or a global solution in terms of the functional residual $\mathbb{E}f(\bar{x}) - f^\star \leq \varepsilon$ (convex case, gradient dominated functions of degree $\alpha = 1$), by different first-order and second-order optimization algorithms. We take into account that the cost of one stochastic Hessian is proportional to $d$ times the cost of the stochastic gradient, where $d$ is the problem dimension, which holds for general dense problems.

| | **Non-convex**, $\varepsilon$-local solution | **Convex** ($\alpha = 1$), $\varepsilon$-global solution | Ref. |
|---|---|---|---|
| Gradient Descent **(GD)** | $n / \varepsilon^2$ | $n / \varepsilon$ | Nesterov (2018) |
| Stochastic Gradient Descent **(SGD)** | $1 / \varepsilon^4$ | $1 / \varepsilon^2$ | Ghadimi & Lan (2013) |
| Stochastic Variance Reduced Grad. **(SVRG)** | $n^{2/3} / \varepsilon^2$ | $n^{2/3} / \varepsilon$ | Johnson & Zhang (2013) |
| Cubic Newton **(CN)** | $nd / \varepsilon^{3/2}$ | $nd / \varepsilon^{1/2}$ | Nesterov & Polyak (2006) |
| Stochastic Cubic Newton **(SCN)** | $1 / \varepsilon^{7/2} + d / \varepsilon^{5/2}$ | $1 / \varepsilon^{5/2} + d / \varepsilon^{3/2}$ | Kohler & Lucchi (2017) Nilesh et al. (2018) |
| Variance Reduced Stoch. Cubic Newton **(VRCN)** | $(nd)^{4/5} \wedge (n^{2/3}d + n) / \varepsilon^{3/2}$ | $(nd)^{4/5} \wedge (n^{2/3}d + n) / \varepsilon^{1/2}$ **(new)** | Zhou et al. (2019) Wang et al. (2019) |
| **(new)** Variance Reduced Stoch. CN with Lazy Hessians **(VRCN-Lazy)** | $(nd)^{5/6} \wedge n\sqrt{d} / \varepsilon^{3/2}$ **(new)** | $(nd)^{5/6} \wedge n\sqrt{d} / \varepsilon^{1/2}$ **(new)** | This work |

Table 1: The total number of stochastic gradient computations for solving the problem with $\varepsilon$ accuracy. $n$ is the number of functions (the data size), and $d$ is the dimension of the problem. We use $x \wedge y$ to denote $\min(x, y)$.

We see that for $d \geq n^{2/3}$ (large dimension setting) it is better to use the new VRCN-Lazy method than the VRCN algorithm. Moreover, note that both in the SCN and VRCN algorithms, we need to solve a cubic subproblem with a new approximate Hessian matrix at *each iteration*. This means that, in case of the exact steps, an expensive matrix factorization needs to be computed every iteration for these algorithms. At the same time, our new VRCN-Lazy method benefits from utilizing a matrix factorization for many steps, significantly improving the total arithmetical cost of the method.

## 2 Notation and Assumptions

We consider the finite-sum optimization problem (1) and we assume that our objective $f$ is bounded from below, denoting $f^\star := \inf_{\boldsymbol{x}} f(\boldsymbol{x})$, and use the following notation: $F_0 := f(\boldsymbol{x}_0) - f^\star$, for some initial $\boldsymbol{x}_0 \in \mathbb{R}^d$. We denote by $\|\boldsymbol{x}\| := \langle \boldsymbol{x}, \boldsymbol{x} \rangle^{1/2}$, $\boldsymbol{x} \in \mathbb{R}^d$, the standard Euclidean norm for vectors, and the spectral norm for symmetric matrices by $\|\boldsymbol{H}\| := \max\{\lambda_{\max}(\boldsymbol{H}), -\lambda_{\min}(\boldsymbol{H})\}$, where $\boldsymbol{H} = \boldsymbol{H}^\top \in \mathbb{R}^{d \times d}$. We will also use $x \wedge y$ to denote $\min(x, y)$.

Throughout this work, we make the following smoothness assumption on the objective $f$:

---

**Assumption 2.1** (Lipschitz Hessian)**.** *The Hessian of $f$ is Lipschitz continuous, for some $L > 0$:*

$$\|\nabla^2 f(\boldsymbol{x}) - \nabla^2 f(\boldsymbol{y})\| \quad \leq \quad L\|\boldsymbol{x} - \boldsymbol{y}\|, \ \forall \boldsymbol{x}, \boldsymbol{y} \in \mathbb{R}^d$$

---

Our goal is to explore the potential of using the Cubically regularized Newton methods to solve problem (1). At each iteration, being at a point $\boldsymbol{x} \in \mathbb{R}^d$, we compute the next point $\boldsymbol{x}^+$ by solving the subproblem of the form

$$\boldsymbol{x}^+ \in \arg\min_{\boldsymbol{y} \in \mathbb{R}^d} \Big\{ \Omega_{M,\boldsymbol{g},\boldsymbol{H}}(\boldsymbol{y}, \boldsymbol{x}) := \langle \boldsymbol{g}, \boldsymbol{y} - \boldsymbol{x} \rangle + \frac{1}{2} \langle \boldsymbol{H}(\boldsymbol{y} - \boldsymbol{x}), \boldsymbol{y} - \boldsymbol{x} \rangle + \frac{M}{6} \|\boldsymbol{y} - \boldsymbol{x}\|^3 \Big\}. \tag{2}$$

Here, $\boldsymbol{g}$ and $\boldsymbol{H}$ are estimates of the gradient $\nabla f(\boldsymbol{x})$ and the Hessian $\nabla^2 f(\boldsymbol{x})$, respectively. Note that solving (2) can be done efficiently even for non-convex problems (see Conn et al. (2000); Nesterov & Polyak (2006); Cartis et al. (2011a)). Generally, the cost of computing $\boldsymbol{x}^+$ is $\mathcal{O}(d^3)$ arithmetic operations, which are needed for evaluating an appropriate factorization of $\boldsymbol{H}$. Hence, it is of a similar order as the cost of the classical Newton's step. Inexact first-order solvers for the cubic subproblem were studied in Carmon & Duchi (2020); Nesterov (2022).

We will be interested to find a second-order stationary point to (1). We call $(\varepsilon, c)$-*approximate second-order local minimum* a point $\boldsymbol{x}$ that satisfies:

$$\|\nabla f(\boldsymbol{x})\| \quad \leq \quad \varepsilon \quad \text{and} \quad \lambda_{min}(\nabla^2 f(\boldsymbol{x})) \quad \geq \quad -c\sqrt{\varepsilon},$$

where $\varepsilon, c > 0$ are given tolerance parameters. Let us define the following accuracy measure (see Nesterov & Polyak (2006)):

$$\mu_c(\boldsymbol{x}) := \max\Big( \|\nabla f(\boldsymbol{x})\|^{3/2}, \frac{-\lambda_{min}(\nabla^2 f(\boldsymbol{x}))^3}{c^{3/2}} \Big).$$

Note that this definition implies that if $\mu_c(\boldsymbol{x}) \leq \varepsilon^{3/2}$ then $\boldsymbol{x}$ is an $(\varepsilon, c)$-approximate local minimum.

**Computing gradients and Hessians.** It is clear that computing the Hessian matrix can be much more expensive than computing the gradient vector. We denote the corresponding arithmetic complexities of computing one Hessian and gradient of $f_i(\boldsymbol{x}), 1 \leq i \leq n$ by *HessCost* and *GradCost*. We will make and follow the convention that $HessCost = d \times GradCost$, where $d$ is the dimension of the problem. For example, this is known to hold for neural networks using the backpropagation algorithm (Kelley, 1960). However, if the Hessian has a sparse structure, the cost of computing the Hessian can be cheaper Nocedal & Wright (2006). Then, we can replace $d$ with the *effective dimension* $d_{\text{eff}} := \frac{HessCost}{GradCost} \leq d$.

## 3 Second-Order Optimization with Helper Functions

In this section, we extend the helper framework previously introduced in (Chayti & Karimireddy, 2022) for first-order optimization methods to second-order optimization.

**General principle.** The general idea is the following: imagine that, besides the objective function $f$, we have access to a helper function $h$ that we think is similar in some sense (that we will define later) to $f$ and thus it should help to minimize it.

Note that many optimization algorithms can be framed in the following sequential way. For a current state $\boldsymbol{x}$, we compute the next state $\boldsymbol{x}^+$ as:

$$\boldsymbol{x}^+ \quad \in \quad \underset{\boldsymbol{y}\in\mathbb{R}^d}{\arg\min}\Big\{ \hat{f}_{\boldsymbol{x}}(\boldsymbol{y}) + Mr_{\boldsymbol{x}}(\boldsymbol{y}) \Big\},$$

where $\hat{f}_{\boldsymbol{x}}(\cdot)$ is an approximation of $f$ around current point $\boldsymbol{x}$, and $r_{\boldsymbol{x}}(\boldsymbol{y})$ is a regularizer that encodes how accurate the approximation is, and $M > 0$ is a regularization parameter. In this work, we are interested in cubically regularized second-order models of the form (2) and we use $r_{\boldsymbol{x}}(\boldsymbol{y}) := \frac{1}{6}\|\boldsymbol{y} - \boldsymbol{x}\|^3$.

Now, let us look at how we can use a helper $h$ to construct the approximation $\hat{f}$. We notice that we can write

$$f(\boldsymbol{y}) \quad := \quad \underbrace{h(\boldsymbol{y})}_{\text{cheap}} + \underbrace{f(\boldsymbol{y}) - h(\boldsymbol{y})}_{\text{expensive}}$$

We discuss the actual practical choices of the helper function $h$ below. We assume now that we can afford the second-order approximation for the cheap part $h$ around the current point $\boldsymbol{x}$. However, approximating the part $f - h$ can be expensive (as for example when the number of elements $n$ in finite sum (1) is huge), or even impossible (due to lack of data). Thus, we would prefer to approximate the expensive part less frequently. For this reason, let us introduce an extra *snapshot point* $\hat{\boldsymbol{x}}$ that is updated less often than $\boldsymbol{x}$. Then, we use it to approximate $f - h$. Another question we still need to ask is *what order should we use to approximate* $f - h$? We will see that order 0 (i.e., a constant) leads us to the basic stochastic methods, while for orders 1 and 2, our methods are akin to classical variance reduction techniques.

Combining the two approximations for $h$ and $f - h$ we get the following model of our objective $f$:

$$\hat{f}_{\boldsymbol{x},\tilde{\boldsymbol{x}}}(\boldsymbol{y}) = C(\boldsymbol{x},\tilde{\boldsymbol{x}}) + \langle \mathcal{G}(h,\boldsymbol{x},\tilde{\boldsymbol{x}}), \boldsymbol{y} - \boldsymbol{x}\rangle + \frac{1}{2}\langle \mathcal{H}(h,\boldsymbol{x},\tilde{\boldsymbol{x}})(\boldsymbol{y} - \boldsymbol{x}), \boldsymbol{y} - \boldsymbol{x}\rangle, \tag{3}$$

where $C(\boldsymbol{x},\tilde{\boldsymbol{x}})$ is a constant, $\mathcal{G}(h,\boldsymbol{x},\tilde{\boldsymbol{x}})$ is a linear term, and $\mathcal{H}(h,\boldsymbol{x},\tilde{\boldsymbol{x}})$ is a matrix. Note that if $\tilde{\boldsymbol{x}} \equiv \boldsymbol{x}$, then the best second-order model of the form (3) is the Taylor polynomial of degree two for $f$ around $\boldsymbol{x}$, and that would yield the exact Newton-type method. However, when the points $\boldsymbol{x}$ and $\tilde{\boldsymbol{x}}$ are different, we obtain much more freedom in constructing our models.

For using this model in our cubically regularized method (2), we only need to define the gradient $\boldsymbol{g} = \mathcal{G}(h,\boldsymbol{x},\tilde{\boldsymbol{x}})$ and the Hessian estimates $\boldsymbol{H} = \mathcal{H}(h,\boldsymbol{x},\tilde{\boldsymbol{x}})$, and we can also treat them differently (using two different helpers, $h_1$ and $h_2$, correspondingly). Thus, we come to the following general second-order (meta)algorithm. We perform $S$ rounds, the length of each round is $m \geq 1$, which is our key parameter:

---

**Algorithm 1** Cubic Newton with helper functions

---

**Require:** $\boldsymbol{x}_0 \in \mathbb{R}^d$, $S$, $m \geq 1$, $M > 0$.

1: **for** $t = 0, \ldots, Sm - 1$ **do**
2:     **if** $t \bmod m = 0$ **then**
3:         Update $\tilde{\boldsymbol{x}}_t$ (using previous states $\boldsymbol{x}_{i\leq t}$)
4:     **else**
5:         $\tilde{\boldsymbol{x}}_t = \tilde{\boldsymbol{x}}_{t-1}$
6:     Form helper functions $h_1, h_2$
7:     Compute the gradient $\boldsymbol{g}_t = \mathcal{G}(h_1, \boldsymbol{x}_t, \tilde{\boldsymbol{x}}_t)$, and the Hessian $\boldsymbol{H}_t = \mathcal{H}(h_2, \boldsymbol{x}_t, \tilde{\boldsymbol{x}}_t)$
8:     Compute the cubic step $\boldsymbol{x}_{t+1} \in \arg\min_{\boldsymbol{y}\in\mathbb{R}^d} \Omega_{M,\boldsymbol{g}_t,\boldsymbol{H}_t}(\boldsymbol{y},\boldsymbol{x}_t)$
   **return** $\boldsymbol{x}_{out}$ using the history $(\boldsymbol{x}_i)_{0\leq i\leq Sm}$

---

In Algorithm 1 we update the snapshot $\tilde{\boldsymbol{x}}$ regularly every $m$ iterations. The two possible options are

$$\tilde{\boldsymbol{x}}_t \quad = \quad \boldsymbol{x}_{t\bmod m} \qquad\qquad \text{(use the last iterate)} \tag{4}$$

or

$$\tilde{\boldsymbol{x}}_t \quad = \quad \arg\min_{i\in\{t-m+1,\ldots,t\}} f(\boldsymbol{x}_i) \qquad \text{(use the best iterate)}. \tag{5}$$

Clearly, option (5) is available only in case we can efficiently estimate the function values. However, we will see that it serves us with better global convergence guarantees for the gradient-dominated functions. It remains to specify how we choose the helpers $h_1$ and $h_2$. We need to assume that they are somehow similar to $f$. Let us present several efficient choices that lead to implementable second-order schemes.

### 3.1 Basic Stochastic Methods

If the objective function $f$ is very "expensive" (for example of the form (1) with $n \to \infty$), one option is to ignore the part $f - h$ i.e. to approximate it by a zeroth-order approximation: $f(\boldsymbol{y}) - h(\boldsymbol{y}) \approx f(\tilde{\boldsymbol{x}}) - h(\tilde{\boldsymbol{x}})$. Since it is a constant, we do not need to update $\tilde{\boldsymbol{x}}$. In this case, we have:

$$\mathcal{G}(h_1, \boldsymbol{x}, \tilde{\boldsymbol{x}}) := \nabla h_1(\boldsymbol{x}), \quad \mathcal{H}(h_2, \boldsymbol{x}, \tilde{\boldsymbol{x}}) := \nabla^2 h_2(\boldsymbol{x}). \tag{6}$$

To treat this choice of the helpers, we assume the following similarity assumptions, which is motivated by the form of the errors for one cubic step (see Theorem C.1):

> **Assumption 3.1** (Bounded similarity). *For $\mathcal{G}, \mathcal{H}$ defined in (6), there exists $\delta_1, \delta_2 \geq 0$ such that it holds, $\forall \boldsymbol{x}, \tilde{\boldsymbol{x}} \in \mathbb{R}^d$:*
> $$\mathbb{E}_{h_1}[\|\mathcal{G}(h_1, \boldsymbol{x}, \tilde{\boldsymbol{x}}) - \nabla f(\boldsymbol{x})\|^{3/2}] \leq \delta_1^{3/2},$$
> $$\mathbb{E}_{h_2}[\|\mathcal{H}(h_2, \boldsymbol{x}, \tilde{\boldsymbol{x}}) - \nabla^2 f(\boldsymbol{x})\|^3] \leq \delta_2^3.$$

*Remark* 3.2. Introducing the random variable $\boldsymbol{\eta} := \mathcal{G}(h_1, \boldsymbol{x}, \tilde{\boldsymbol{x}}) - \nabla f(\boldsymbol{x})$, the first condition in the assumption can be rewritten as a bound for its $3/2$-moment:

$$\mathbb{E}_{h_1}[\|\boldsymbol{\eta}\|^{3/2}] \leq \delta_1^{3/2}. \tag{7}$$

Note that the standard assumption in the literature on first-order methods (e.g. Lan (2020)) is the boundedness of the second moment (or the *bounded variance*):

$$\mathbb{E}_{h_1}[\|\boldsymbol{\eta}\|^2] \leq \sigma^2, \tag{8}$$

which is stronger than (7). Indeed, using Jensen's inequality for the expectation as applied to a concave function $g(t) = t^{3/4}, t \geq 0$, we obtain that

$$\mathbb{E}_{h_1}[\|\boldsymbol{\eta}\|^{3/2}] = \mathbb{E}_{h_1}[(\|\boldsymbol{\eta}\|^2)^{3/4}] \leq \left(\mathbb{E}_{h_1}[\|\boldsymbol{\eta}\|^2]\right)^{3/4} \overset{(8)}{\leq} \sigma^{3/2}.$$

Thus, (7) holds with $\delta_1 := \sigma$. At the same time, for the Hessian approximation, we assume a stronger guarantee of the boundedness of the *third moment*. Note that if both $f$ and $h_2$ have $\beta$-Lipschitz continuous gradient, we immediately obtain the deterministic upper bound: $\delta_2 \leq 2\beta$.

Under Assumption 3.1, we prove the following theorem:

> **Theorem 3.3.** *Under Assumptions 2.1 and 3.1, and $M \geq L$, for an output of Algorithm 1 $\boldsymbol{x}_{out}$ chosen uniformly at random from $(\boldsymbol{x}_i)_{0 \leq i \leq Sm}$, we have:*
> $$\mathbb{E}[\mu_M(\boldsymbol{x}_{out})] = \mathcal{O}\left(\frac{\sqrt{M}F_0}{Sm} + \frac{\delta_2^3}{M^{3/2}} + \delta_1^{3/2}\right).$$

We see that according to this result, we can get $\mathbb{E}[\mu_M(\boldsymbol{x}_{out})] \leq \varepsilon^{3/2}$ only for $\varepsilon > \delta_1$. In other words, we can converge only to a certain *neighborhood around a stationary point*, determined by the error $\delta_1$ of the stochastic gradients.

However, as we will show next, this seemingly pessimistic dependence leads to the same rate of classical subsampled Cubic Newton methods discovered in Kohler & Lucchi (2017); Xu et al. (2017; 2016).

At this point, let us discuss the specific case of *stochastic optimization*, where $f$ has the form of (1), with $n$ potentially being very large. In this case, it is customary to sample batches at random and assume the noise to be bounded in expectation. Precisely speaking, if we assume the standard assumption that for one index sampled uniformly at random, we have $\mathbb{E}_i\|\nabla f(\boldsymbol{x}) - \nabla f_i(\boldsymbol{x})\|^2 \leq \sigma_g^2$ and $\mathbb{E}_i\|\nabla^2 f(\boldsymbol{x}) - \nabla^2 f_i(\boldsymbol{x})\|^3 \leq \sigma_h^3$, $\forall \boldsymbol{x}$, then it is possible to show that for

$$h_1 = \frac{1}{b_g}\sum_{i \in \mathcal{B}_g} f_i \text{ and } h_2 = \frac{1}{b_h}\sum_{i \in \mathcal{B}_h} f_i, \tag{9}$$

where batches $\mathcal{B}_g, \mathcal{B}_h \subseteq [n]$ sampled uniformly at random and of sizes $b_g$ and $b_h$ respectively, Assumption 3.1 is satisfied with (see, e.g., Tropp et al. (2015)): $\delta_1 = \frac{\sigma_g}{\sqrt{b_g}}$ and $\delta_2 = \tilde{\mathcal{O}}(\frac{\sigma_h}{\sqrt{b_h}})$. Note that according to this result, we can use *the same random subsets* of indices $\mathcal{B}_g, \mathcal{B}_h$ for all iterations of the method.

**Corollary 3.4.** *In Algorithm 1, let us choose $M = L$ and $m = 1$, with basic helpers (9). Then, according to Theorem 3.3, for any $\varepsilon > 0$, to reach an $(\varepsilon, L)$-approximate second-order local minimum, we need at most $S = \frac{\sqrt{L}F_0}{\varepsilon^{3/2}}$ iterations with $b_g = \left(\frac{\sigma_g}{\varepsilon}\right)^2$ and $b_h = \frac{\sigma_h^2}{\varepsilon}$. Therefore, the total complexity of the method in terms of the gradient oracle calls is*

$$\mathcal{O}\left(\frac{\sigma_g^2}{\varepsilon^{7/2}} + \frac{\sigma_h^2}{\varepsilon^{5/2}}d_{\text{eff}}\right) \times GradCost. \tag{10}$$

Recall that $d_{\text{eff}} \leq d$. Bound (10) improves upon the complexity $\mathcal{O}(\frac{1}{\varepsilon^4}) \times GradCost$ of the first-order SGD for non-convex optimization Ghadimi & Lan (2013), unless $d_{\text{eff}} > \frac{1}{\varepsilon^{3/2}}$ (high cost of computing the Hessians).

## 3.2 Let the Objective Guide Us

If the objective $f$ is such that we can afford to access its gradients and Hessians from time to time (functions of the form (1) with $n < \infty$ being "reasonable"), then we can do better than the previous chapter. In this case, we can use a better approximation of the term $f(\boldsymbol{y}) - h(\boldsymbol{y})$. From a theoretical point of view, we can treat the case when $f$ is only differentiable once, and thus, we can only use a first-order approximation of $f - h$; in this case, we will only be using the Hessian of the helper $h$ but only gradients of $f$. However, in our case, if we assume we have access to gradients, then we can also have access to the Hessians of $f$ as well (from time to time); for this reason, we consider a second-order approximation of the term $f - h$. If we follow the procedure that we described above, we find:

$$\mathcal{G}(h_1, \boldsymbol{x}, \tilde{\boldsymbol{x}}) := \nabla h_1(\boldsymbol{x}) - \nabla h_1(\tilde{\boldsymbol{x}}) + \nabla f(\tilde{\boldsymbol{x}}) + (\nabla^2 f(\tilde{\boldsymbol{x}}) - \nabla^2 h_1(\tilde{\boldsymbol{x}}))(\boldsymbol{x} - \tilde{\boldsymbol{x}}), \tag{11}$$

$$\mathcal{H}(h_2, \boldsymbol{x}, \tilde{\boldsymbol{x}}) := \nabla^2 h_2(\boldsymbol{x}) - \nabla^2 h_2(\tilde{\boldsymbol{x}}) + \nabla^2 f(\tilde{\boldsymbol{x}}). \tag{12}$$

We see that there is an explicit dependence on the snapshot $\tilde{\boldsymbol{x}}$, and thus, we need to address the question of how we should update this snapshot point in Algorithm 1. In general, we can update it with a certain probability $p \sim \frac{1}{m}$, or we can use more advanced combinations of past iterates (e.g., a moving average). However, for simplicity, we study option 4 (i.e., using the last iterate for updating the snapshot $\tilde{\boldsymbol{x}}$); thus, it is updated only once every $m$ iterations.

We also need to address the question of measuring the similarity in this case. Since we employ a second-order approximation of $f - h$, it seems natural to compare the function $f$ and its helpers $h_1$ and $h_2$ by using the difference between their third derivatives or, equivalently, the Hessian Lipschitz constant of their difference. Precisely, we make the following similarity assumption:

**Assumption 3.5** (Lipschitz similarity). *For $\mathcal{G}, \mathcal{H}$ defined in (11) and (12), there exists $\delta_1, \delta_2 \geq 0$ such that it holds, $\forall \boldsymbol{x}, \tilde{\boldsymbol{x}} \in \mathbb{R}^d$:*

$$\mathbb{E}_{h_1}[\|\mathcal{G}(h_1, \boldsymbol{x}, \tilde{\boldsymbol{x}}) - \nabla f(\boldsymbol{x})\|^{3/2}] \leq \delta_1^{3/2}\|\boldsymbol{x} - \tilde{\boldsymbol{x}}\|^3,$$

$$\mathbb{E}_{h_2}[\|\mathcal{H}(h_2, \boldsymbol{x}, \tilde{\boldsymbol{x}}) - \nabla^2 f(\boldsymbol{x})\|^3] \leq \delta_2^3\|\boldsymbol{x} - \tilde{\boldsymbol{x}}\|^3.$$

In particular, if $f - h_1$ and $f - h_2$ have Lipschitz Hessians (with constants $\delta_1$ and $\delta_2$ respectively) then $h_1$ and $h_2$ satisfy Assumption 3.5.

Under this assumption, we show that the errors resulting from the use of the snapshot can be successfully balanced by choosing $M$ satisfying:

$$4\left(\frac{\delta_1}{M}\right)^{3/2} + 73\left(\frac{\delta_2}{M}\right)^3 \leq \frac{1}{24m^3}. \tag{13}$$

And we have the following theorem:

**Theorem 3.6.** *Let $f, h_1, h_2$ verify Assumptions 2.1,3.5, and let the regularization parameter $M$ is chosen such that $M \geq L$ and (13) is satisfied. Then, for the output $\boldsymbol{x}_{out}$ of Algorithm 1 chosen uniformly at random from $(\boldsymbol{x}_i)_{0 \leq i \leq Sm:=T}$, we have:*

$$\mathbb{E}[\mu_M(\boldsymbol{x}_{out})] = \mathcal{O}\left(\frac{\sqrt{M}F_0}{Sm}\right).$$

In particular, we can choose $M = \max(L, 32\delta_1 m^2, 16\delta_2 m)$ which gives

$$\mathbb{E}[\mu_M(\boldsymbol{x}_{out})] = \mathcal{O}\left(\frac{\sqrt{L}F_0}{Sm} + \frac{\sqrt{\delta_2}F_0}{S\sqrt{m}} + \frac{\sqrt{\delta_1}F_0}{S}\right). \tag{14}$$

Based on the choices of the helpers $h_1$ and $h_2$, we can have many algorithms. We discuss particular implementations in the following sections.

We start by presenting the *variance reduction* combined with the *Lazy Hessian* updates, which rely on sampling batches randomly. Our new method will significantly improve complexity bound (10) obtained for the basic helpers, and achieve the best overall performance among all known variants of stochastic second-order methods (see Table 1).

Then we discuss other applications: *the core sets*, which try to intelligently find a weighted representative batch describing the whole dataset; *semi-supervised learning*, engineering the helpers using unlabeled data; and, more generally, *auxiliary learning*, which tries to leverage auxiliary tasks in training a given main task. We show that the auxiliary tasks can naturally be treated as helpers.

### 3.3 Variance Reduction and Lazy Hessians

First, note that choosing $h_1 = h_2 = f$ gives the classical Cubic Newton method (Nesterov & Polyak, 2006), whereas choosing $h_1 = f$ and $h_2 = 0$ gives the Lazy Cubic Newton (Doikov et al., 2022). In both cases, we recuperate the known rates of convergence. However, these choices requires to compute the full gradients, that can be expensive for solving problem (1) with large number $n$ of components.

The following lemma demonstrates that we can create helper functions $h$ with lower similarity to the main function $f$ by employing sampling and averaging.

**Lemma 3.7.** *Let $f = \frac{1}{n}\sum_{i=1}^{n} f_i$ such that all $f_i$ are twice differentiable and have $L$-Lipschitz Hessians. Let $\mathcal{B} \subset \{1, \cdots, n\}$ be of size $b$ and sampled with replacement uniformly at random, and define $h_{\mathcal{B}} = \frac{1}{b}\sum_{i \in \mathcal{B}} f_i$, then $h_{\mathcal{B}}$ satisfies Assumption 3.5 with $\delta_1 = \frac{L}{\sqrt{b}}$ and $\delta_2 = \mathcal{O}(\frac{\sqrt{\log(d)}L}{\sqrt{b}})$.*

**Choice of the parameter $m$ in Algorithm 1.** Minimizing the total arithmetic cost, we will set

$$m = \arg\min\left\{ \text{Grad}(m, \varepsilon) + d \cdot \text{Hess}(m, \varepsilon) \right\}, \tag{15}$$

where $\text{Grad}(m, \varepsilon)$ and $\text{Hess}(m, \varepsilon)$ denote the number of gradients and Hessians required to find an $\varepsilon$ stationary point. Now, we are ready to discuss several particular cases that are direct consequences from Theorem 3.6 and Lemma 3.7.

**General variance reduction.** If we sample batches $\mathcal{B}_g$ and $\mathcal{B}_h$ of sizes $b_g$ and $b_h$ consecutively at random and choose, as in Section 3.1:

$$h_1 = \frac{1}{b_g}\sum_{i \in \mathcal{B}_g} f_i \qquad \text{and} \qquad h_2 = \frac{1}{b_h}\sum_{i \in \mathcal{B}_h} f_i, \tag{16}$$

and use these helpers along with the estimates (11), (12), we obtain the *Variance Reduced Cubic Newton* algorithm (Zhou et al., 2019; Wang et al., 2019). According to Lemma 3.7, this choice corresponds to $\delta_1 = L/\sqrt{b_g}$ and $\delta_2 = \tilde{\mathcal{O}}(L/\sqrt{b_h})$. For $b_g \sim m^4 \wedge n, b_h \sim m^2 \wedge n$ and $M = L$, we obtain the non-convex convergence rate of the order: $\mathbb{E}[\mu_L(\boldsymbol{x}_{out})] = \mathcal{O}\left(\frac{\sqrt{L}F_0}{Sm}\right)$. This is the same rate as of the full Cubic Newton. However, our cost per iteration is much smaller due to using stochastic oracles. Minimizing the total

arithmetic cost, we choose $m$ according to (15), as to minimize the following expression: $g^{VR}(n,d) := \min_m\{\frac{dn+d(m^3 \wedge nm)+(m^5 \wedge nm)}{m}\}$. Then we reach an $(\varepsilon, L)$-approximate second-order local minimum in at most

$$\mathcal{O}(\frac{g^{VR}(n,d)}{\varepsilon^{3/2}}) \times GradCost \tag{17}$$

gradient oracle calls.

**Variance reduction with Lazy Hessians.** We can also use lazy updates for Hessians combined with variance-reduced gradients; this corresponds to choosing

$$\boxed{h_1 = \frac{1}{b_g}\sum_{i \in \mathcal{B}_g} f_i \qquad \text{and} \qquad h_2 = 0,} \tag{18}$$

which implies (according to Lemma 3.7) that $\delta_1 = L/\sqrt{b_g}$ and $\delta_2 = L$. In this case, we need $b_g \sim m^2$ to obtain a convergence rate: $\mathbb{E}[\mu_L(\boldsymbol{x}_{out})] = \mathcal{O}(\frac{\sqrt{L}F_0}{S\sqrt{m}})$, which matches the convergence rate of the full version of the Lazy Cubic Newton method (Doikov et al. (2022)), while in our method we use stochastic gradients. In this case, the choice of $m$ minimizes the following expression: $g^{Lazy}(n,d) := \min_m\{\frac{nd+(m^3 \wedge mn)}{\sqrt{m}}\}$. Then we guarantee to reach an $(\varepsilon, mL)$-approximate second-order local minimum in at most

$$\mathcal{O}(\frac{g^{Lazy}(n,d)}{\varepsilon^{3/2}}) \times GradCost \tag{19}$$

gradient oracle calls.

**To be lazy or not to be?** We can show that $g^{Lazy}(n,d) \sim (nd)^{5/6} \wedge n\sqrt{d}$ and $g^{VR}(n,d) \sim (nd)^{4/5} \wedge (n^{2/3}d+n)$. In particular, for $d \geq n^{2/3}$ we have $g^{Lazy}(n,d) \leq g^{VR}(n,d)$ and thus for $\boxed{d \geq n^{2/3}}$ *it is better to use Lazy Hessians along with the variance reduction*, obtaining complexity (19) than becomes strictly better than (17). We also note that for the Lazy approach, we can keep a factorization of the Hessian (this factorization induces most of the cost of solving the cubic subproblem); thus, it is as if we only need to solve the subproblem once every $m$ iterations, so the Lazy approach has a big advantage compared to the general approach, and the advantage becomes even bigger for the case of large dimensions.

Note that according to our theory, we could use the same random batches $\mathcal{B}_g, \mathcal{B}_h \subseteq [n]$ *generated once* for all iterations. However, using the resampled batches can lead to a more stable convergence.

### 3.4 Other Applications

The result in (14) is general enough to include many other applications that are limited only by our imagination; there are, to cite a few such applications :

**Core sets.** (Bachem et al., 2017) The idea of core sets is simple: can we summarize a potentially large data set using only a few (potentially weighted) important examples? Many reasons, such as redundancy, make the answer yes. Devising approaches to find such core sets is outside of the scope of this work. Formally, for the objective function $f(\boldsymbol{x})$ of the finite-sum structure (1), the core set helper can be given by the following form:

$$h(\boldsymbol{x}) = \frac{1}{n}\sum_{j=1}^{m} w_j \bar{f}_j(\boldsymbol{x}), \tag{20}$$

where each $\bar{f}_j, 1 \leq j \leq m$ is a core representative (or cluster) for a given dataset.

**Example 3.8.** *Let core representatives $\bar{f}_1, \ldots, \bar{f}_m$ be fixed, and assume that each function from our target problem* (1) *be given by*

$$f_i(\boldsymbol{x}) := \bar{f}_{j_i}(\boldsymbol{x}) + \varepsilon_i(\boldsymbol{x}), \quad 1 \leq i \leq n,$$

*where $j_i \in \{1, \ldots, m\}$ is the index of the corresponding core representative and $\varepsilon_i(\boldsymbol{x})$ is some noise. Then, function of from* (20) *with weights $w_j := \sum_{i=1}^{n}[j_i = j]$ will be a helper function according to our framework. Assuming that each $\varepsilon_i$ has Lipschitz Hessian with constant $L_\varepsilon$ will imply Assumption 3.5 for $h_1 = h_2 = h$ with $\delta_1 = \delta_2 = L_\varepsilon$.*

**Reusing batches during training.** In general, we can see from (14) that if we have batches $\mathcal{B}_g, \mathcal{B}_h$ such that they are $\delta_1$ and $\delta_2$ similar to $f$ respectively, then we can keep reusing the same batch $\mathcal{B}_g$ for at least $\sqrt{\frac{L}{\delta_1}}$ times, and $\mathcal{B}_h$ for $\frac{L}{\delta_2}$ all the while guaranteeing an improved rate. So then, if we can design such small batches with small $\delta_1$ and $\delta_2$, we can keep reusing them and enjoy the improved rate without needing large batches.

**Auxiliary learning.** (Baifeng et al.; Aviv et al.; Xingyu et al.) study how a given task $f$ can be trained in the presence of auxiliary (related) tasks; our approach can indeed be used for auxiliary learning by treating the auxiliaries as helpers; if we compare (14) to the rate that we obtained without the use of the helpers: $\mathcal{O}(\frac{\sqrt{L}F_0}{S})$, we see that we have a better rate using the helpers/auxiliary tasks when $\frac{1}{m} + \frac{\sqrt{\delta_2}}{\sqrt{mL}} + \frac{\sqrt{\delta_1}}{\sqrt{L}} \leq 1$.

**Semi-supervised learning.** (Yang et al., 2021) Semi-supervised learning is a machine learning approach that uses both labeled and unlabeled data during training. In general, we can use the unlabeled data to construct the helpers; we can start, for example, by using random labels for the helpers and improving the labels with training. There are at least two special cases where our theory implies improvement by only assigning random labels to the unlabeled data. In fact, for both regularized least squares and logistic regression, we notice that the Hessian is independent of the labels (only depends on inputs).

Indeed, let us consider the classical logistic regression model (e.g., Murphy (2012)) for the set of labeled data $\{(\boldsymbol{a}_i, b_i)\}_{i=1}^n$, where $b_i = \pm 1$. Then fitting the model can be formulated as the following optimization problem:

$$\min_{\boldsymbol{x} \in \mathbb{R}^d} \left[ f(\boldsymbol{x}) \quad := \quad \frac{1}{n} \sum_{i=1}^n \ell(-y_i \boldsymbol{a}_i^\top \boldsymbol{x}) \right], \tag{21}$$

where $\ell(t) := \ln(1 + e^t)$. Note that $\ell'(t) = \sigma(t) := \frac{1}{1+e^{-t}}$ and $\ell''(t) = \sigma(t) \cdot \sigma(-t)$. Hence, the second derivative of the logistic loss is an *even function*. Computing the Hessian of $f$ directly, we obtain

$$\nabla^2 f(\boldsymbol{x}) \quad = \quad \frac{1}{n} \sum_{i=1}^n \ell''(-b_i \boldsymbol{a}_i^\top \boldsymbol{x})(-b_i \boldsymbol{a}_i)(-b_i \boldsymbol{a}_i)^\top \quad = \quad \frac{1}{n} \sum_{i=1}^n \ell''(\boldsymbol{a}_i^\top \boldsymbol{x}) \boldsymbol{a}_i \boldsymbol{a}_i^\top.$$

Thus, we see that the last expression *does not depend on the input labels* $b_i$. Therefore, if the unlabeled data comes from the same distribution as the labeled data, then we can use it to construct helpers which, at least theoretically, have $\delta_1 = \delta_2 = 0$. Because the Hessian is independent of the labels, we can technically endow the unlabeled data with random labels. Theorem 3.6 will imply in this case $\mathbb{E}[\mu_L(\boldsymbol{x}_{out})] = \mathcal{O}(\frac{\sqrt{L}F_0}{Sm})$, where $S$ is the number of times we use labeled data and $S(m-1)$ is the number of unlabeled data.

**Example 3.9.** *Consider the following helper,*

$$h(\boldsymbol{x}) \quad := \quad \frac{1}{m} \sum_{j=1}^m \ell(-\bar{y}_j \boldsymbol{b}_j^\top \boldsymbol{x}), \tag{22}$$

*where labels $\bar{y}_j = \pm 1$ are randomly generated, while data $\{\boldsymbol{b}_j\}_{j=1}^m$ has the same distribution as the target data $\{\boldsymbol{a}_i\}_{i=1}^n$ from (21). Then, taking into account that the Hessian of each $f_i(\boldsymbol{x}) = \ell(-y_i \boldsymbol{a}_i^\top \boldsymbol{x})$ is Lipschitz with constant $L = \max_i \|\boldsymbol{a}_i\|^3 / (6\sqrt{3})$, we obtain that using helper (22) as $h_1 = h_2 = h$ provides us with the guarantees required by Assumption 3.5 with $\delta_1 = L/\sqrt{m}$ and $\delta_2 = \tilde{\mathcal{O}}(L/\sqrt{m})$ (see also Lemma 3.7).*

See Figure 2 for the empirical validation of this approach.

# 4 Gradient Dominated Functions

In this section, we consider the class of gradient-dominated functions defined below.

**Assumption 4.1.** $(\tau, \alpha)$-*gradient dominated.* *A function $f$ is called gradient dominated on set if it holds, for some $\alpha \geq 1$ and $\tau > 0$:*

$$f(\boldsymbol{x}) - f^\star \leq \tau \|\nabla f(\boldsymbol{x})\|^\alpha, \qquad \forall \boldsymbol{x} \in \mathbb{R}^d. \tag{23}$$

Examples of functions satisfying this assumption are convex functions ($\alpha = 1$) and strongly convex functions ($\alpha = 2$); see Appendix F.1. For such functions, we can guarantee convergence (in expectation) to a *global minimum*, i.e., we can find a point $\boldsymbol{x}$ such that $f(\boldsymbol{x}) - f^\star \leq \varepsilon$.

The gradient dominance property is interesting because many non-convex functions have been shown to satisfy it (uanzhi & Yang, 2017; Hardt & Ma, 2016; Masiha et al., 2022). Furthermore, besides convergence to a global minimum, we get improved rates of convergence.

We note that for $\alpha > 3/2$ (and only for this case), we also need to assume the following (stronger) inequality:

$$\mathbb{E}f(\boldsymbol{x}_t) - f^\star \quad \leq \quad \tau\mathbb{E}\big[\|\nabla f(\boldsymbol{x}_t)\|\big]^\alpha, \tag{24}$$

where the expectation is taken with respect to the iterates $(\boldsymbol{x}_t)$ of our algorithms; this is a stronger assumption than (23). To avoid using (24), we can assume that the iterates belong to some compact set $Q \subset \mathbb{R}^d$ and that the gradient norm is uniformly bounded: $\forall \boldsymbol{x} \in Q : \|\nabla f(\boldsymbol{x})\| \leq G$. Then, a $(\tau, \alpha)$-gradient dominated on set $Q$ function is also a $(\tau G^{\alpha-3/2}, 3/2)$-gradient dominated on this set for any $\alpha > 3/2$.

In the following theorem, we extend the results of Theorem 3.3 to gradient-dominated functions, for the basic stochastic helpers:

---

**Theorem 4.2.** *Under Assumptions 2.1,3.1,4.1, for $M \geq L$ and $T := Sm$, we have*
*- For $1 \leq \alpha < 3/2$:*

$$\mathbb{E}[f(\boldsymbol{x}_T)] - f^\star \quad = \quad \mathcal{O}\Big(\big(\tfrac{\alpha\sqrt{M}\tau^{3/(2\alpha)}}{(3-2\alpha)T}\big)^{\frac{2\alpha}{3-2\alpha}} + \tau\tfrac{\delta_2^{2\alpha}}{M^\alpha} + \tau\delta_1^\alpha\Big). \tag{25}$$

*- For $\alpha = 3/2$:*

$$\mathbb{E}[f(\boldsymbol{x}_T)] - f^\star \quad = \quad \mathcal{O}\Big(F_0 \exp\big(\tfrac{-T}{1+\sqrt{M}\tau}\big) + \tau\tfrac{\delta_2^3}{M^\alpha} + \tau\delta_1^{3/2}\Big). \tag{26}$$

*- For $3/2 < \alpha \leq 2$, let $h_0 = \mathcal{O}(F_0/(\sqrt{M}\tau^{\frac{3}{2\alpha}})^{\frac{2\alpha}{3-2\alpha}})$. Then for $T \geq t_0 = \mathcal{O}(h_0^{\frac{2\alpha-3}{2\alpha}}\log(h_0))$ we have:*

$$\mathbb{E}[f(\boldsymbol{x}_T)] - f^\star \quad = \quad \mathcal{O}\Big((\sqrt{M}\tau^{\frac{3}{2\alpha}})^{\frac{2\alpha}{3-2\alpha}}\big(\tfrac{1}{2}\big)^{(\frac{2\alpha}{3})^{T-t_0}} + \tau\tfrac{\delta_2^{2\alpha}}{M^\alpha} + \tau\delta_1^\alpha\Big). \tag{27}$$

---

Theorem 4.2 shows (up to the noise level) the global sublinear rate for $1 \leq \alpha < 3/2$, the global linear rate for $\alpha = 3/2$ (which can also be seen by taking the limit $\alpha \to 3/2$ in (25)) and the superlinear rate for $\alpha > 3/2$, after the initial phase of $t_0$ iterations.

We would like to point that instead of Assumption 3.1 we only need to assume $\mathbb{E}_{h_1}[\|\nabla h_1(\boldsymbol{x}) - \nabla f(\boldsymbol{x})\|^\alpha] \leq \delta_1^\alpha$ and $\mathbb{E}_{h_2}[\|\nabla^2 h_2(\boldsymbol{x}) - \nabla^2 f(\boldsymbol{x})\|^{2\alpha}] \leq \delta_2^{2\alpha}$ which might be weaker depending on the value of $\alpha$. We also prove the following theorem, which extends the results in Theorem 3.6 to the gradient-dominated functions, for the advanced helpers. In this case, we set the snapshot line 3 in Algorithm 1) as in (5) i.e., the snapshot corresponds to the state with the smallest value of $f$ during the last $m$ iterations.

---

**Theorem 4.3.** *Under Assumptions 2.1,3.5,4.1, for $M = \max(L, 34\delta_1 m^2, 11\delta_2 m)$, we have:*
*- For $1 \leq \alpha < 3/2$:*

$$\mathbb{E}[f(\boldsymbol{x}_{Sm})] - f^\star \quad = \quad \mathcal{O}\Big(\big(\tfrac{\alpha\sqrt{M}\tau^{3/(2\alpha)}}{(3-2\alpha)Sm}\big)^{\frac{2\alpha}{3-2\alpha}}\Big).$$

*- For $\alpha = 3/2$:*

$$\mathbb{E}[f(\boldsymbol{x}_{Sm})] - f^\star \quad = \quad \mathcal{O}\Big(F_0\big(1 + \tfrac{m}{\sqrt{M}\tau}\big)^{-S}\Big).$$

*- For $3/2 < \alpha \leq 2$, let $h_0 = \mathcal{O}(\tfrac{F_0}{(\frac{\sqrt{M}}{m}\tau^{\frac{3}{2\alpha}})^{\frac{2\alpha}{3-2\alpha}}})$. Then for $S \geq s_0 = \mathcal{O}(h_0^{\frac{2\alpha-3}{2\alpha}}\log(h_0))$ we have:*

$$\mathbb{E}[f(\boldsymbol{x}_{Sm})] - f^\star \quad = \quad \Big((\tfrac{\sqrt{M}}{m}\tau^{\frac{3}{2\alpha}})^{\frac{2\alpha}{3-2\alpha}}\big(\tfrac{1}{2}\big)^{(\frac{2\alpha}{3})^{S-s_0}}\Big)$$

---

Again, the same behavior is observed as for Theorem 4.2, but this time without noise (variance reduction is working). To the best of our knowledge, this is the first time such an analysis has been made. As a direct consequence of our results, we obtain new global complexities for the variance-reduced and lazy variance-reduced Cubic Newton methods on the classes of gradient-dominated functions. Note that in the simplest case of the deterministic Lazy Cubic Newton ($h_1 = f$ and $h_2 = 0$), we enhance the complexity results from (Doikov et al., 2022) to the classes of convex and strongly convex functions, establishing faster rates of convergence.

Let us compare the statements of Theorems 4.2 and 4.3, for convex functions ($\alpha = 1$). Theorem 4.2 guarantees convergence to a $\varepsilon-$global minimum in at most $\mathcal{O}(\frac{1}{\varepsilon^{5/2}} + \frac{d}{\varepsilon^{3/2}}) \times GradCost$ gradient computations, while Theorem 4.3 only needs $\mathcal{O}(\frac{g(n,d)}{\sqrt{\varepsilon}}) \times GradCost$, where $g(n,d)$ is either $g^{Lazy}(n,d) = (nd)^{5/6} \wedge n\sqrt{d}$ or $g^{VR}(n,d) = (nd)^{4/5} \wedge (n^{2/3}d + n)$. See Appendix F.3 for more details.

## 5 Experiments

### 5.1 To Be Lazy or Not to Be

To verify our findings from Subsection 3.3, we consider a logistic regression problem (21) with $\ell_2$-regularization on the "a9a" data set Chang & Lin (2011). We consider the variance-reduced cubic Newton method from (Zhou et al., 2019) (referred to as "full VR"), its lazy version where we do not update the snapshot Hessian ("Lazy VR"), the stochastic Cubic Newton method ("SCN"), the Cubic Newton algorithm ("CN"), Gradient Descent with line search ("GD") and Stochastic Gradient Descent ("SGD"). We report the results in terms of time and gradient arithmetic computations needed to arrive at a given level of convergence.

Figure 1 shows that the lazy version saves both time and arithmetic computations without sacrificing the convergence precision. In these graphs, *Gradcost* is computed using the convention that computing one hessian is $d$ times as expensive as computing one gradient.

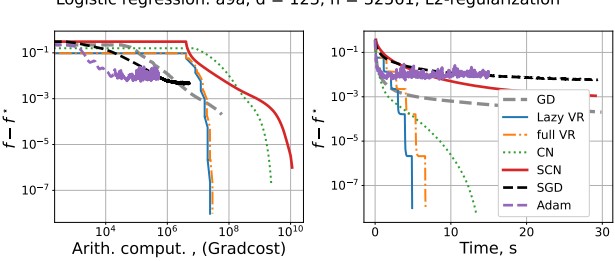

Figure 1: Comparison of the convergence of different algorithms. We see that "Lazy VR" has the same convergence speed as its full version "full VR" and the cubic Newton method "CN", while it needs less time and fewer arithmetic computations.

### 5.2 Auxiliary Learning

Our goal is to show that the helper framework is very general and that it goes beyond the variance reduction and lazy Hessian computations. For the previously considered problem of training the logistic regression (using the same "a9a" data set), we suppose that we also have access to unlabeled data (in this sense, this becomes semi-supervised learning). Specifically, we have a labeled dataset $\mathcal{D}_l = \{(\boldsymbol{a}_i, b_i)\}_{i=1}^{N_l}$ and an unlabeled data set $\mathcal{D}_u = \{\boldsymbol{a}_i\}_{i=N_l+1}^{N_l+N_u}$, we suppose that both data sets are sampled from the same distribution $\mathcal{P}_{(\boldsymbol{a},\boldsymbol{b})}$. Our goal is to minimize

$$f(\boldsymbol{x}) = \mathbb{E}_{(\boldsymbol{a},\boldsymbol{b}) \sim \mathcal{P}_{(\boldsymbol{a},\boldsymbol{b})}}[\log(1 + \exp(-b\boldsymbol{x}^\top \boldsymbol{a}))].$$

A simple computation (see Section 3.4) shows that the Hessian of $f$ only depends on $\mathcal{P}_{\boldsymbol{a}}$, and, for this reason, we can use unlabeled data to construct a good approximation of the true Hessian (if we can sample from $\mathcal{P}_{\boldsymbol{a}}$,

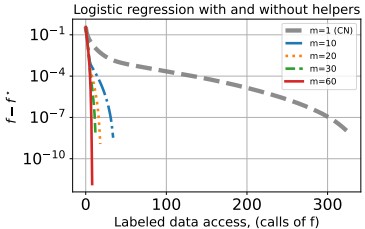

Figure 2: Cubic Newton method with and without using the helper function $h$. For $m = 1$, this is simply the classic Cubic Newton method. To give an intuitive meaning to the plot, $\frac{1}{m}$ is the percentage of labeled data used during training. We can clearly see that using our approach, we benefit a lot from the helper function $h$.

we construct the exact Hessian and thus have a helper $h$ with $\delta_1 = \delta_2 = 0$). Let

$$h(\boldsymbol{x}) = \mathbb{E}_{\boldsymbol{a} \sim \mathcal{P}_{\boldsymbol{a}}, \boldsymbol{b} \sim Random\{\pm 1\}}[\log(1 + \exp(-b\boldsymbol{x}^\top \boldsymbol{a}))],$$

where $Random\{\pm 1\}$ is any distribution on labels. In our experiments, we use uniform distribution. Figure 2 shows that, indeed, we can benefit a lot from using this helper function. We note that this observed benefit comes at the cost of performing more steps using gradients and Hessians of the helper function $h$.

## 5.3 Non-convex experiments

We go back to comparing the algorithms in 5.1. We consider, now, non-convex problems. First we consider the logistic regression with a non-convex regularizer $Reg(\boldsymbol{x}) = \sum_{i=1}^{d} \frac{x_i^2}{1+x_i^2}$ (Kohler & Lucchi, 2017). Thus, we minimize

$$f(\boldsymbol{x}) \quad = \quad \frac{1}{n} \sum_{i=1}^{n} \log(1 + \exp(-b_i \boldsymbol{x}^\top \boldsymbol{a}_i)) + \lambda Reg(\boldsymbol{x}).$$

Figure 3 shows the results in this case. Again, we see that "lazy VR" reduces both time and gradient equivalent computations without sacrificing the convergence speed.

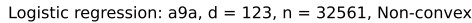
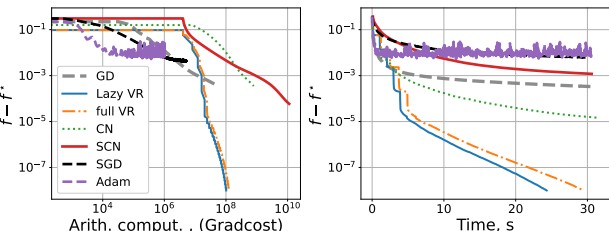

Figure 3: Comparison of the convergence of different algorithms. We see that using our approach, we benefit a lot from the helper function $h$.

Second, we consider a simple diagonal neural network with L2 loss with data generated from a normal distribution. specifically, we want to minimize

$$f(\boldsymbol{x} := (\boldsymbol{u}, \boldsymbol{v})) = \frac{1}{n} \sum_{i=1}^{n} \|\boldsymbol{a}_i^\top \boldsymbol{u} \circ \boldsymbol{v} - b_i\|^2 + \frac{\lambda}{2} \|\boldsymbol{x}\|^2 \,,$$

where $\circ$ is the element-wise vector product.

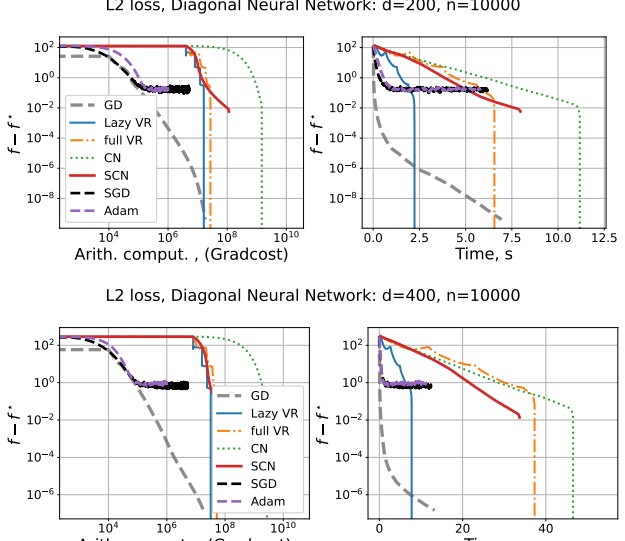

Figure 5: Effect of increasing the dimension on the convergence of the different optimization algorithms we consider. We notice that with increased dimension, the gap between our method "Lazy VR" and "full VR" widens meaning "lazy VR" saves more time as the dimension of the problem grows.

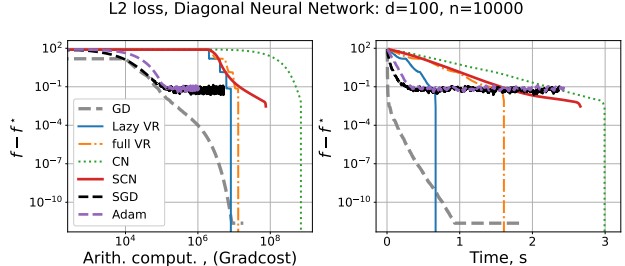

Figure 4: Comparison of the convergence of the different algorithms. Except for gradient descent ("GD"), which performs very well in this case, again, the same conclusions as in Figure 2 with respect to "Lazy VR" can be said.

Figure 4 shows that compared to other second-order methods, "Lazy VR" has considerable time and computation savings. It also performs closely to gradient descent with line search, which performs very well in this case. Figure 5 shows the same experiment for larger dimensions, most importantly we see that the gap between our "Lazy VR" and the "full VR" method grows with $d$, this is in accord with our theory which predicts an increased advantage of "lazy VR" as the dimension grows.

## 5.4 Hessian Cost vs Gradient Cost

We consider again a diagonal neural network and estimate the time costs needed for computing its gradient, Hessian, decomposing the Hessian, and solving the cubic subproblem. Figure 6 shows that the average cost of computing the Hessian is significantly higher than the cost of computing one gradient, and the quotient grows approximately linearly with d (the dimension of the problem). Figure 6 also shows that the cost of decomposing the Hessian dominates that of solving the cubic subproblem which explains how our lazy VR method saves time compared to other methods.

## 5.5 Additional experiments

We consider, in this section, other datasets from the LibSVM library (Chang & Lin, 2011).

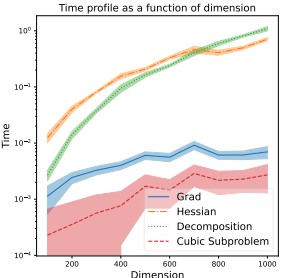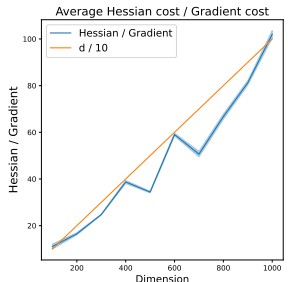

Figure 6: **(left)** times needed to compute the gradient, the Hessian, decompose the Hessian and solve the cubic subproblem for a diagonal neural network with $n = 10000$ and different values of the dimension $d$. **(right)** average time for computing the Hessian divided by the average time needed to compute the gradient. we notice that the Hessian computation cost is almost proportional to $d\times$ the gradient cost. We also see that the time needed to decompose the Hessian dominates the time needed to solve the cubic subproblem.

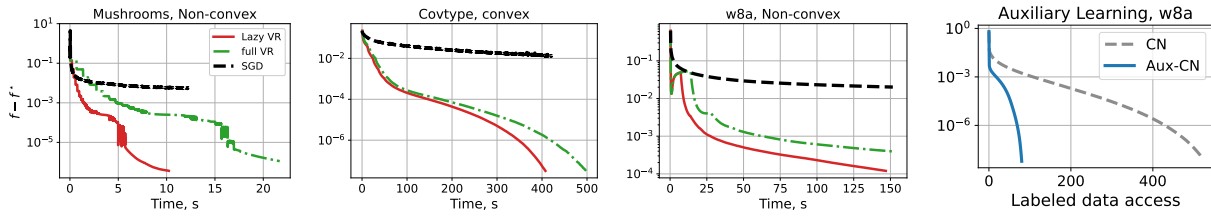

Figure 7: Experiments on Mushrooms ($n = 8124, d = 112$), Covtype ($n = 581012, d = 54$), w8a ($n = 49749, d = 300$) datasets.

Figure 7 clearly shows how our method outperforms the baselines by saving time (or calls to the main function in the case of auxiliary learning) without sacrificing performance.

## 6 Limitations and possible extensions

**Estimating similarity between the helpers and the main function.** While we show in this work that we can have an improvement over training alone, this supposes that we know the similarity constants $\delta_1, \delta_2$; hence, it will be interesting to have approaches that can adapt to such constants.

**Engineering helper functions.** Building helper task with small similarities is also an interesting idea. Besides the examples in supervised learning and core-sets that we provide, it is not evident how to do it in a generalized way.

**Using the helper to regularize the cubic subproblem.** We note that while we proposed to approximate the "cheap" part as well in Section 3, one other theoretically viable approach is to keep it intact and approximately solve a "proximal type" problem involving $h$; this will lead to replacing $L$ by $\delta$, but the subproblem is even more difficult to solve. However, our theory suggests that we don't need to solve this subproblem exactly; we only need $m \geq \frac{L}{\delta}$; we do not treat this case here.

## 7 Conclusion

In this work, we proposed a general theory for using stochastic and auxiliary information in the context of the Cubically regularized Newton method. Our theory encapsulates the classical stochastic methods, as well as Variance Reduction and the methods with the Lazy Hessian updates.

Our new methods posses the best-known global complexity bounds for stochastic second-order optimization in the non-convex case, and significantly benefit in terms of total arithmetic computations, by reducing the number of computed Hessians and the required number of matrix factorizations.

For auxiliary learning, we demonstrate a provable benefit of using auxiliary data compared to training alone. Besides investigating general non-convex functions for which we proved global convergence rates to a second-order stationary point, we also studied the classes of gradient-dominated functions with improved rates of convergence to the global minima.

## Acknowledgements

This work was supported by the Swiss State Secretariat for Education, Research and Innovation (SERI) under contract number 22.00133.

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

## A Reproducibility

Our code is available with all the details necessary for reproducing our results in `https://github.com/elmahdichayti/Unified-Convergence-Theory-of-Cubic-Newton-s-method`.

## B Additional Experiments

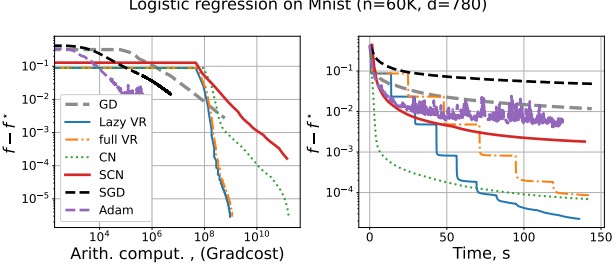

Figure 8: Logistic regression on the MNIST dataset for different algorithms. We notice that second-order methods clearly outperform first-order methods. We also note how cubic Newton (CN) starts as the fastest but our Lazy VR method catches up to it.

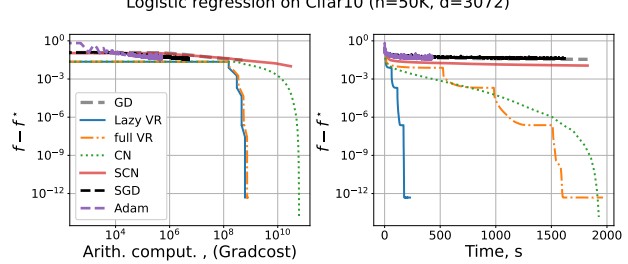

Figure 9: Logistic regression on the CIFAR10 dataset for different algorithms. We notice that second-order methods clearly outperform first-order methods. We also note that our Lazy VR has a big advantage in terms of time compared to all other methods.

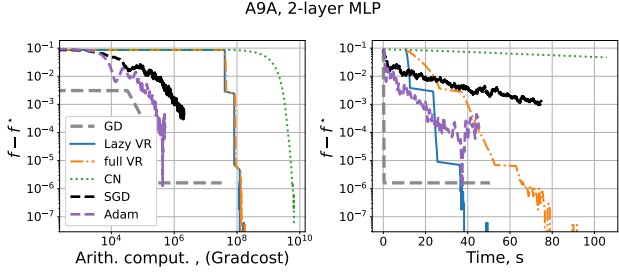

Figure 10: A two-layer NN trained on the A9A dataset. We notice that GD (with line search) outperforms all the algorithms but gets stuck at an error of $10^{-6}$. Lazy VR is the second fastest and even ends up beating GD to a smaller error, suggesting that it escaped the local minimum GD was stuck at. Again Lazy VR saves more time compared to full VR.

## C   Theoretical Preliminaries

We consider the general problem

$$\min_{\boldsymbol{x}\in\mathbb{R}^d} f(\boldsymbol{x})$$

Where $f$ is twice differentiable with $L$-Lipschitz Hessian i.e.:

$$\|\nabla^2 f(\boldsymbol{x}) - \nabla^2 f(\boldsymbol{y})\| \le L\|\boldsymbol{x}-\boldsymbol{y}\|, \qquad \forall \boldsymbol{x},\boldsymbol{y}\in\mathbb{R}^d. \tag{28}$$

As a direct consequence of (28) (see (Nesterov & Polyak, 2006; Nesterov, 2018)) we have for all $\boldsymbol{x},\boldsymbol{y}\in\mathbb{R}^d$:

$$\|\nabla f(\boldsymbol{y}) - \nabla f(\boldsymbol{x}) - \nabla^2 f(\boldsymbol{x})(\boldsymbol{y}-\boldsymbol{x})\| \le \frac{L}{2}\|\boldsymbol{x}-\boldsymbol{y}\|^2, \tag{29}$$

$$|f(\boldsymbol{y}) - f(\boldsymbol{x}) - \langle\nabla f(\boldsymbol{x}),\boldsymbol{y}-\boldsymbol{x}\rangle - \frac{1}{2}\langle\nabla^2 f(\boldsymbol{x})(\boldsymbol{y}-\boldsymbol{x}),\boldsymbol{y}-\boldsymbol{x}\rangle| \le \frac{L}{6}\|\boldsymbol{y}-\boldsymbol{x}\|^3. \tag{30}$$

For $\boldsymbol{x}$ and $\boldsymbol{x}^+$ defined as in Equation (2) i.e.

$$\boldsymbol{x}^+ \in \arg\min_{\boldsymbol{y}\in\mathbb{R}^d}\left\{ \Omega_{M,\boldsymbol{g},\boldsymbol{H}}(\boldsymbol{y},\boldsymbol{x}) := \langle \boldsymbol{g}, \boldsymbol{y}-\boldsymbol{x}\rangle + \tfrac{1}{2}\langle \boldsymbol{H}(\boldsymbol{y}-\boldsymbol{x}),\boldsymbol{y}-\boldsymbol{x}\rangle + \tfrac{M}{6}\|\boldsymbol{y}-\boldsymbol{x}\|^3 \right\}. \tag{31}$$

The optimality condition of (31) ensures that :

$$\langle \boldsymbol{g}, \boldsymbol{x}^+ - \boldsymbol{x}\rangle + \langle \boldsymbol{H}(\boldsymbol{x}^+-\boldsymbol{x}),\boldsymbol{x}^+-\boldsymbol{x}\rangle + \frac{M}{2}r^3 = 0, \tag{32}$$

where we denoted $r = \|\boldsymbol{x}^+ - \boldsymbol{x}\|$.

It is also known that the solution to (31) verifies:

$$\boldsymbol{H} + \frac{M}{2}r\mathbb{I} \succeq 0 \tag{33}$$

We start by proving the following Theorem

---

**Theorem C.1.** *For any $\boldsymbol{x}\in\mathbb{R}^d$, let $\boldsymbol{x}^+$ be defined by (2). Then, for $M\ge L$ we have:*

$$f(\boldsymbol{x}) - f(\boldsymbol{x}^+) \ge \frac{1}{1008\sqrt{M}}\mu_M(\boldsymbol{x}^+) + \frac{M\|\boldsymbol{x}-\boldsymbol{x}^+\|^3}{72} - \frac{4\|\nabla f(\boldsymbol{x})-\boldsymbol{g}\|^{3/2}}{\sqrt{M}} - \frac{73\|\nabla^2 f(\boldsymbol{x})-\boldsymbol{H}\|^3}{M^2}.$$

---

Using (30) with $\boldsymbol{y} = \boldsymbol{x}^+$ and $\boldsymbol{x} = \boldsymbol{x}$ and for $M \ge L$ we have:

$$f(\boldsymbol{x}^+) \overset{(30)}{\le} f(\boldsymbol{x}) + \langle\nabla f(\boldsymbol{x}),\boldsymbol{x}^+-\boldsymbol{x}\rangle + \tfrac{1}{2}\langle\nabla^2 f(\boldsymbol{x})(\boldsymbol{x}^+-\boldsymbol{x}),\boldsymbol{x}^+-\boldsymbol{x}\rangle + \tfrac{L}{6}r^3$$

$$\overset{(32)+(33)}{\le} f(\boldsymbol{x}) - \tfrac{6M-4L}{24}r^3 + \langle\nabla f(\boldsymbol{x})-\boldsymbol{g},\boldsymbol{x}^+-\boldsymbol{x}\rangle$$

$$+ \tfrac{1}{2}\langle(\nabla^2 f(\boldsymbol{x})-\boldsymbol{H})(\boldsymbol{x}^+-\boldsymbol{x}),\boldsymbol{x}^+-\boldsymbol{x}\rangle$$

$$\overset{M\ge L}{\le} f(\boldsymbol{x}) - \tfrac{M}{12}r^3 + \langle\nabla f(\boldsymbol{x})-\boldsymbol{g},\boldsymbol{x}^+-\boldsymbol{x}\rangle$$

$$+ \tfrac{1}{2}\langle(\nabla^2 f(\boldsymbol{x})-\boldsymbol{H})(\boldsymbol{x}^+-\boldsymbol{x}),\boldsymbol{x}^+-\boldsymbol{x}\rangle.$$

Using Young's inequality $xy \le \frac{x^p}{p} + \frac{y^q}{q}$ $\forall x, y \in \mathbb{R}$ $\forall p, q > 1$ s.t $\frac{1}{p} + \frac{1}{q} = 1$ we have:

$$\langle\nabla f(\boldsymbol{x})-\boldsymbol{g},\boldsymbol{x}^+-\boldsymbol{x}\rangle \le \frac{M}{36}r^3 + \frac{2\sqrt{12}}{3\sqrt{M}}\|\nabla f(\boldsymbol{x})-\boldsymbol{g}\|^{3/2},$$

and

$$\frac{1}{2}\langle(\nabla^2 f(\boldsymbol{x})-\boldsymbol{H})(\boldsymbol{x}^+-\boldsymbol{x}),\boldsymbol{x}^+-\boldsymbol{x}\rangle \le \frac{M}{36}r^3 + \frac{72}{M^2}\|\nabla^2 f(\boldsymbol{x})-\boldsymbol{H}\|^3.$$

Mixing all these ingredients, we get

**Lemma C.2.** *For any $M \geq L$, it holds*

$$f(\boldsymbol{x}) - f(\boldsymbol{x}^+) \geq \frac{M}{36} r^3 - \frac{3}{\sqrt{M}} \|\nabla f(\boldsymbol{x}) - \boldsymbol{g}\|^{3/2} - \frac{72}{M^2} \|\nabla^2 f(\boldsymbol{x}) - \boldsymbol{H}\|^3. \tag{34}$$

Using (29) we have:

$$\|\nabla f(\boldsymbol{x}^+) - \boldsymbol{g} - \boldsymbol{H}(\boldsymbol{x}^+ - \boldsymbol{x}) + \boldsymbol{g} - \nabla f(\boldsymbol{x}) + (\boldsymbol{H} - \nabla^2 f(\boldsymbol{x}))(\boldsymbol{x}^+ - \boldsymbol{x})\| \leq \frac{L}{2} r^2,$$

applying the triangular inequality we get for $M \geq L$ :

$$\begin{aligned}
\|\nabla f(\boldsymbol{x}^+)\| &\leq \frac{L}{2} r^2 + \|\boldsymbol{g} + \boldsymbol{H}(\boldsymbol{x}^+ - \boldsymbol{x})\| + \|\nabla f(\boldsymbol{x}) - \boldsymbol{g}\| + \|\nabla^2 f(\boldsymbol{x}) - \boldsymbol{H}\| r \\
&\leq \frac{L+2M}{2} r^2 + \|\nabla f(\boldsymbol{x}) - \boldsymbol{g}\| + \frac{1}{2M} \|\nabla^2 f(\boldsymbol{x}) - \boldsymbol{H}\|^2 \\
&\leq \frac{3M}{2} r^2 + \|\nabla f(\boldsymbol{x}) - \boldsymbol{g}\| + \frac{1}{2M} \|\nabla^2 f(\boldsymbol{x}) - \boldsymbol{H}\|^2.
\end{aligned}$$

By the convexity of $x \mapsto x^{3/2}$ we have for any $(a_i) \geq 0$ : $(\sum_i a_i x_i)^{3/2} \leq (\sum_i a_i)^{1/2} \sum_i a_i x_i^{3/2}$, applying this to the above inequality we get

**Lemma C.3.** *For any $M \geq L$, it holds*

$$\frac{1}{\sqrt{M}} \|\nabla f(\boldsymbol{x}^+)\|^{3/2} \leq 3M r^3 + \frac{2}{\sqrt{M}} \|\nabla f(\boldsymbol{x}) - \boldsymbol{g}\|^{3/2} + \frac{1}{M^2} \|\nabla^2 f(\boldsymbol{x}) - \boldsymbol{H}\|^3 \tag{35}$$

We can also bound the smallest eigenvalue of the Hessian. Using the smoothness of the Hessian we have:

$$\begin{aligned}
\nabla^2 f(\boldsymbol{x}^+) &\succeq \nabla^2 f(\boldsymbol{x}) - L\|\boldsymbol{x}^+ - \boldsymbol{x}\| \mathbb{I} \\
&\succeq \boldsymbol{H} + \nabla^2 f(\boldsymbol{x}) - \boldsymbol{H} - Lr\mathbb{I} \\
&\succeq \boldsymbol{H} - \|\nabla^2 f(\boldsymbol{x}) - \boldsymbol{H}\| \mathbb{I} - Lr\mathbb{I} \\
&\overset{(33)}{\succeq} -\frac{Mr}{2} \mathbb{I} - \|\nabla^2 f(\boldsymbol{x}) - \boldsymbol{H}\| \mathbb{I} - Lr\mathbb{I}.
\end{aligned}$$

Which means for $M \geq L$ we have:

$$-\lambda_{min}(\nabla^2 f(\boldsymbol{x}^+)) \leq \frac{3Mr}{2} + \|\nabla^2 f(\boldsymbol{x}) - \boldsymbol{H}\|.$$

Then the convexity of $x \mapsto x^3$ leads to the following lemma:

**Lemma C.4.** *For any $M \geq L$, it holds*

$$\frac{-\lambda_{min}(\nabla^2 f(\boldsymbol{x}^+))^3}{M^2} \leq 14M r^3 + \frac{4}{M^2} \|\nabla^2 f(\boldsymbol{x}) - \boldsymbol{H}\|^3 \tag{36}$$

Now the quantity $\mu_M(\boldsymbol{x}) = \max(\|\nabla f(\boldsymbol{x})\|^{3/2}, \frac{-\lambda_{min}(\nabla^2 f(\boldsymbol{x}^+))^3}{M^{3/2}})$ which we can be bounded using Lemmas C.3 and C.4:

$$\frac{1}{\sqrt{M}} \mu(\boldsymbol{x}^+) \leq 14M r^3 + \frac{2}{\sqrt{M}} \|\nabla f(\boldsymbol{x}) - \boldsymbol{g}\|^{3/2} + \frac{4}{M^2} \|\nabla^2 f(\boldsymbol{x}) - \boldsymbol{H}\|^3. \tag{37}$$

Combining Lemma C.2 and (37) we get the inequality given in Theorem C.1:

$$f(\boldsymbol{x}) - f(\boldsymbol{x}^+) \geq \frac{1}{1008\sqrt{M}} \mu_M(\boldsymbol{x}^+) + \frac{M}{72} r^3 - \frac{4}{\sqrt{M}} \|\nabla f(\boldsymbol{x}) - \boldsymbol{g}\|^{3/2} - \frac{73}{M^2} \|\nabla^2 f(\boldsymbol{x}) - \boldsymbol{H}\|^3.$$

$\square$

# D More on Section 3.1

## D.1 Better Similarity using Sampling

One common approach for constructing gradient and Hessian estimates is sub-sampling. The idea behind sub-sampling is simple: for an objective of the form in (1), we randomly sample two batches $\mathcal{B}_g$ and $\mathcal{B}_h$ of sizes $b_g$ and $b_h$ consecutively from the distribution $\mathcal{D}$ and define:

$$\boldsymbol{g}_{t,\mathcal{B}_g} = \frac{1}{b_g} \sum_{i \in \mathcal{B}_g} \nabla f_i(\boldsymbol{x}_t) \quad \text{and} \quad \boldsymbol{H}_{t,\mathcal{B}_h} = \frac{1}{b_h} \sum_{i \in \mathcal{B}_h} \nabla^2 f_i(\boldsymbol{x}_t). \tag{38}$$

In this particular scenario, the "elementary" estimates $\nabla f(\boldsymbol{x}_t, \zeta)$ and $\nabla^2 f(\boldsymbol{x}_t, \zeta)$ are unbiased, and we will assume that they satisfy $\mathbb{E}_i \|\nabla f(\boldsymbol{x}) - \nabla f_i(\boldsymbol{x})\|^2 \le \sigma_g^2$ and $\mathbb{E}_i \|\nabla^2 f(\boldsymbol{x}) - \nabla^2 f_i(\boldsymbol{x})\|^3 \le \sigma_h^3$.

**Lemma D.1.** *For the estimators defined in* (38) *we have*[1]:

$$\mathbb{E}\|\nabla f(\boldsymbol{x}_t) - \boldsymbol{g}_{t,\mathcal{B}_g}\|^2 \le \frac{\sigma_g^2}{b_g} \quad \text{and} \quad \mathbb{E}\|\nabla^2 f(\boldsymbol{x}_t) - \boldsymbol{H}_{t,\mathcal{B}_h}\|^3 \le \mathcal{O}\left(\log(d)^{3/2} \frac{\sigma_h^3}{b_h^{3/2}}\right).$$

Lemma D.1 demonstrates how the utilization of batching can decrease the noise. To simplify things, we can keep in mind this straightforward rule:

If we employ a batch of size $b_a$, then we need to modify $\sigma_a$ by $\frac{\sigma_a}{\sqrt{b_a}}$ for $a \in \{g, h\}$.

Lemma D.1 is based on the following two Lemmas:

**Lemma D.2.** *(Lyapunov's inequality) For any random variable $X$ and any $0 < s < t$ we have*

$$\mathbb{E}[|X|^s]^{1/s} \le \mathbb{E}[|X|^t]^{1/t}.$$

and

**Lemma D.3.** *Suppose that $q \ge 2$, $p \ge 2$, and fix $r \ge \max(q, 2\log(p))$. Consider i.i.d. random self-adjoint matrices $Y_1, \cdots, Y_N$ with dimension $p \times p$, $\mathbb{E}[Y_i] = 0$. It holds that:*

$$\left[\mathbb{E}[\|\sum_{i=1}^N Y_i\|_2^q]\right]^{1/q} \le 2\sqrt{er}\|\left(\sum_{i=1}^N \mathbb{E}[Y_i^2]\right)^{1/2}\|_2 + 4er\mathbb{E}[\max_i \|Y_i\|_2^q]^{1/q}.$$

Lemma D.3 can be found in (Zhou et al., 2019).

Now if we have $X_1, \cdots, X_b \in \mathbb{R}^d$, $b$ i.i.d vector-valued random variables such that $\mathbb{E}[X_i] = \mu$ and $\mathbb{E}[\|X_i - \mu\|^2] \le \sigma^2$ then by applying Lemma D.2 we get:

$$\mathbb{E}[\|\frac{1}{b}\sum_i X_i - \mu\|^{3/2}] \quad \le \quad \mathbb{E}[\|\frac{1}{b}\sum_i X_i - \mu\|^2]^{3/4} \quad \le \quad \frac{\sigma^{3/2}}{b^{3/4}}.$$

When we have $b$ i.i.d matrix-valued random variables $Y_1, \cdots, Y_b \in \mathbb{R}^{d \times d}$ such that $\mathbb{E}[Y_i] = \mu$, $\mathbb{E}[\|Y_i - \mu\|^2] \le \sigma_2^2$ and $\mathbb{E}[\|Y_i - \mu\|^3] \le \sigma_3^3$ (by Jensen's inequality $\sigma_2 \le \sigma_3$), then by applying Lemma D.3 we get:

$$\mathbb{E}[\|\frac{1}{b}\sum_i Y_i - \mu\|^3] \quad \le \quad \left(2\sqrt{\frac{2e\log(d)}{b}}\sigma_2 + \frac{8e\log(d)}{b}\sigma_3\right)^3$$

$$= \quad \mathcal{O}(\frac{\log(d)^{3/2}\sigma_3^3}{b^{3/2}})$$

These last two inequalities are identical to the statement of Lemma D.1.

---

[1]Here and everywhere $\mathcal{O}(\cdot)$ hides an absolute numerical constant.

## D.2 Proof of Theorem 3.3

We use here $\delta_1 = \sigma_g$ and $\delta_2 = \sigma_h$.
Combining both Theorem C.1, Assumption 3.1 and Lemma D.2 we get:

$$\mathbb{E}f(\boldsymbol{x}_t) - \mathbb{E}f(\boldsymbol{x}_{t+1}) \quad \geq \quad \frac{1}{1008\sqrt{M}}\mathbb{E}\mu_M(\boldsymbol{x}_{t+1}) - \frac{4}{\sqrt{M}}\mathbb{E}\|\nabla f(\boldsymbol{x}_t) - \boldsymbol{g}_t\|^{3/2}$$

$$- \frac{73}{M^2}\mathbb{E}\|\nabla^2 f(\boldsymbol{x}_t) - \boldsymbol{H}_t\|^3$$

$$\overset{\text{Lemma } D.2}{\geq} \quad \frac{1}{1008\sqrt{M}}\mathbb{E}\mu_M(\boldsymbol{x}_{t+1}) - \frac{4}{\sqrt{M}}\sigma_g^{3/2} - \frac{73}{M^2}\sigma_h^3.$$

By summing the above inequality from $t = 0$ to $t = T - 1$ and rearranging we get:

$$\frac{1}{1008T}\sum_{t=1}^{T}\mathbb{E}\mu_M(\boldsymbol{x}_t) \quad \leq \quad \sqrt{M}\frac{\mathbb{E}f(\boldsymbol{x}_0) - \mathbb{E}f(\boldsymbol{x}_T)}{T} + 4\sigma_g^{3/2} + \frac{73}{M^{3/2}}\sigma_h^3.$$

All is left to is to use the fact that $\mathbb{E}f(\boldsymbol{x}_0) - \mathbb{E}f(\boldsymbol{x}_T) \leq f(\boldsymbol{x}_0) - f^\star = F_0$, and by the definition of $\boldsymbol{x}_{out}$: $\mathbb{E}\mu_M(\boldsymbol{x}_{out}) = \frac{1}{T}\sum_{t=1}^{T}\mathbb{E}\mu_M(\boldsymbol{x}_t)$, thus:

$$\frac{1}{1008}\mathbb{E}\mu_M(\boldsymbol{x}_{out}) \quad \leq \quad \frac{\sqrt{M}F_0}{T} + \frac{73}{M^{3/2}}\sigma_h^3 + 4\sigma_g^{3/2}.$$

$\square$

# E Proofs of Section 3.2

## E.1 Proof of Lemma 3.7

We have

$$f(\boldsymbol{x}) \quad = \quad \frac{1}{n}\sum_{i=1}^{n}f_i(\boldsymbol{x})$$

and we suppose that each $f_i$, $1 \leq i \leq n$ have the $L$-Lipschitz Hessians. Therefore, $f$ also has the $L$-Lipschitz Hessian. Thus, $f_i - f$ has the $2L$-Lipschitz Hessian.

Applying (2.1) and 29 to $f_i - f$ we get

$$\|\mathcal{G}(f_i, \boldsymbol{x}, \tilde{\boldsymbol{x}}) - \nabla f(\boldsymbol{x})\| \leq L\|\boldsymbol{x} - \tilde{\boldsymbol{x}}\|^2$$

and

$$\|\mathcal{H}(f_i, \boldsymbol{x}, \tilde{\boldsymbol{x}}) - \nabla^2 f(\boldsymbol{x})\| \leq 2L\|\boldsymbol{x} - \tilde{\boldsymbol{x}}\|.$$

We note also the if $i$ is chosen at random then $\mathbb{E}_i\mathcal{G}(f_i, \boldsymbol{x}, \tilde{\boldsymbol{x}}) = \nabla f(\boldsymbol{x})$ and $\mathbb{E}_i\mathcal{H}(f_i, \boldsymbol{x}, \tilde{\boldsymbol{x}}) = \nabla^2 f(\boldsymbol{x})$.

By using the properties of variance we have

$$\mathbb{E}_{\mathcal{B}}\|\mathcal{G}(f_{\mathcal{B}}, \boldsymbol{x}, \tilde{\boldsymbol{x}}) - \nabla f(\boldsymbol{x})\|^2 \leq \frac{\mathbb{E}_i\|\mathcal{G}(f_i, \boldsymbol{x}, \tilde{\boldsymbol{x}}) - \nabla f(\boldsymbol{x})\|^2}{b} \leq \frac{L^2}{b}\|\boldsymbol{x} - \tilde{\boldsymbol{x}}\|^4.$$

Applying Lemmas D.2 and D.3 we get

$$\mathbb{E}_{\mathcal{B}}\|\mathcal{G}(f_{\mathcal{B}}, \boldsymbol{x}, \tilde{\boldsymbol{x}}) - \nabla f(\boldsymbol{x})\|^3 \quad \leq \quad \frac{L^3}{b^{3/2}}\|\boldsymbol{x} - \tilde{\boldsymbol{x}}\|^3$$

and

$$\mathbb{E}_{\mathcal{B}}\|\mathcal{H}(f_{\mathcal{B}}, \boldsymbol{x}, \tilde{\boldsymbol{x}}) - \nabla^2 f(\boldsymbol{x})\|^3 \quad \leq \quad \left(2\sqrt{\frac{2e\log(d)}{b}} + \frac{8e\log(d)}{b}\right)^3 \mathbb{E}_i\|\mathcal{H}(f_i, \boldsymbol{x}, \tilde{\boldsymbol{x}}) - \nabla^2 f(\boldsymbol{x})\|^3$$

$$\leq \quad \underbrace{\left(2\sqrt{\frac{2e\log(d)}{b}} + \frac{8e\log(d)}{b}\right)^3}_{\mathcal{O}\left(\left(\frac{\log(d)}{b}\right)^{3/2}\right)} L^3\|\boldsymbol{x} - \tilde{\boldsymbol{x}}\|^3.$$

$\square$

### E.2    Proof of Theorem 3.6

We use Theorem C.1 and denote $r_{i+1} = \|\boldsymbol{x}_{i+1} - \boldsymbol{x}_i\|$. Then by the definition of the similarity between $h_1, h_2$ and $f$, we have:

$$
\mathbb{E}f(\boldsymbol{x}_{sm+i}) - \mathbb{E}f(\boldsymbol{x}_{sm+i+1}) \;\geq\; \tfrac{1}{216\sqrt{M}}\mathbb{E}\mu_M(\boldsymbol{x}_{sm+i+1})
$$
$$
+ \mathbb{E}\Big[\tfrac{M}{72}r_{sm+i+1}^3 - \big(\tfrac{4\delta_1^{3/2}}{\sqrt{M}} + \tfrac{73\delta_2^3}{M^2}\big)\|\boldsymbol{x}_{sm+i} - \boldsymbol{x}_{sm}\|^3\Big].
$$

Summing it for $0 \leq i \leq m-1$, we obtain

$$
\mathbb{E}f(x_{sm}) - \mathbb{E}f(x_{(s+1)m}) \;\geq\; \sum_{i=0}^{m-1} \tfrac{1}{216\sqrt{M}}\mathbb{E}\mu_M(\boldsymbol{x}_{sm+i+1})
$$
$$
+ \mathbb{E}\Big[\tfrac{M}{72}r_{sm+i+1}^3 - \big(\tfrac{4\delta_1^{3/2}}{\sqrt{M}} + \tfrac{73\delta_2^3}{M^2}\big)\|\boldsymbol{x}_{sm+i} - \boldsymbol{x}_{sm}\|^3\Big].
$$

We note that $\|\boldsymbol{x}_{sm+i} - \boldsymbol{x}_{sm}\| \leq \sum_{j=1}^{i-1} r_{sm+j}$. Therefore,

$$
\mathbb{E}f(x_{sm}) - \mathbb{E}f(x_{(s+1)m}) \;\geq\; \sum_{i=0}^{m-1} \tfrac{1}{216\sqrt{M}}\mathbb{E}\mu_M(\boldsymbol{x}_{sm+i+1})
$$
$$
+ \mathbb{E}\Big[\tfrac{M}{72}r_{sm+i+1}^3 - \big(\tfrac{4\delta_1^{3/2}}{\sqrt{M}} + \tfrac{73\delta_2^3}{M^2}\big)(\sum_{j=1}^{i-1} r_{sm+j})^3\Big].
$$

We apply now the following inequality (from (Doikov et al., 2022)):

$$
\sum_{k=1}^{m-1}\Big(\sum_{i=1}^{k} r_i\Big)^3 \leq \tfrac{m^3}{3}\sum_{k=1}^{m-1} r_k^3,
$$

which is true for any positive numbers $\{r_k\}_{k\geq 1}$ and any $m \geq 1$. Hence,

$$
\sum_{i=0}^{m-1}\Big[\tfrac{M}{72}r_{sm+i+1}^3 - \big(\tfrac{4\delta_1^{3/2}}{\sqrt{M}} + \tfrac{73\delta_2^3}{M^2}\big)(\sum_{j=1}^{i-1} r_{sm+j})^3\Big]
$$
$$
\geq \;\; \Big(\tfrac{M}{72} - \tfrac{m^3}{3}\big(\tfrac{4\delta_1^{3/2}}{\sqrt{M}} + \tfrac{73\delta_2^3}{M^2}\big)\Big)\sum_{i=0}^{m-1} r_{sm+i+1}^3.
$$

The above quantity is thus positive if $\tfrac{M}{72} - \tfrac{m^3}{3}\big(\tfrac{4\delta_1^{3/2}}{\sqrt{M}} + \tfrac{73\delta_2^3}{M^2}\big) \geq 0$.

Equivalently, for $M$ satisfying

$$
\boxed{4(\tfrac{\delta_1}{M})^{3/2} + 73(\tfrac{\delta_2}{M})^3 \leq \tfrac{1}{24m^3}} \tag{39}
$$

we have

$$
\mathbb{E}f(x_{sm}) - \mathbb{E}f(x_{(s+1)m}) \;\geq\; \tfrac{m}{216\sqrt{M}}\tfrac{1}{m}\sum_{i=0}^{m-1}\mathbb{E}\mu(x_{sm+i+1}).
$$

Summing it up for $0 \leq s \leq S-1$ gives

$$
\tfrac{\sqrt{M}(f(x_0)-f^\star)}{Sm} \;\geq\; \tfrac{1}{216Sm}\sum_{s=0}^{S-1}\sum_{i=0}^{m-1}\mathbb{E}\mu(x_{sm+i+1}).
$$

And thus by the definition of $\boldsymbol{x}_{out}$ we have:

$$
\mathbb{E}\mu(x_{out}) \;\leq\; 216\tfrac{\sqrt{M}(f(x_0)-f^\star)}{Sm}.
$$

$\square$

# F Gradient dominated functions

## F.1 Examples of gradient dominated functions

Let us provide several main examples of functions satisfying (23):

**Example F.1.** *Let $f$ be convex on a bounded convex set $Q$ of diameter $D$, and let solution $\boldsymbol{x}^\star$ to (1) belong to $Q$. Then, we have:*

$$f(\boldsymbol{x}) - f^\star \quad \leq \quad \langle \nabla f(\boldsymbol{x}), \boldsymbol{x} - \boldsymbol{x}^\star \rangle \quad \leq \quad D\|\nabla f(\boldsymbol{x})\|, \qquad \forall \boldsymbol{x} \in Q.$$

*Therefore, $f$ is $(D, 1)$-gradient dominated.*

**Example F.2.** *Let $f$ be uniformly convex of degree $s \geq 2$ with some constant $\mu > 0$:*

$$f(\boldsymbol{y}) \quad \geq \quad f(\boldsymbol{x}) + \langle \nabla f(\boldsymbol{x}), \boldsymbol{y} - \boldsymbol{x} \rangle + \tfrac{\mu}{s}\|\boldsymbol{y} - \boldsymbol{x}\|^s, \qquad \forall \boldsymbol{x}, \boldsymbol{y} \in \mathbb{R}^d.$$

*Then, $f$ is $\left(\frac{s-1}{s}(\frac{1}{\mu})^{\frac{1}{s-1}}, \frac{s}{s-1}\right)$-gradient dominated (see, e.g. Doikov & Nesterov (2021)).*

The statement of Example F.2 is easily proven by minimizing both sides of the inequality defining uniformly convex functions.

In particular, uniformly convex functions of degree $s = 2$ are known as *strongly convex*, and we see that they satisfy condition (23) with $\tau = \frac{1}{2\mu}$ and $\alpha = 2$. However, the function class (23) is much wider and it includes also some problems with *non-convex objectives* Nesterov & Polyak (2006).

## F.2 Special cases of Theorem 4.2

For **convex functions** (the case $\alpha = 1$) Theorem 4.2 implies that for $M = \max\{L, \frac{\delta_2 T}{2D}\}$ we have the rate

$$\mathbb{E}[f(\boldsymbol{x}_{out})] - f^\star = \mathcal{O}\left(\frac{LD^3}{T^2} + \frac{\delta_2 D^2}{T} + \delta_1 D\right). \tag{40}$$

Equation (40) has been obtained by (Agafonov et al., 2020) using an assumption that the noise is bounded almost surely. Using the gradient and Hessian estimates in (38), for $\varepsilon > 0$ and $M = L$, to reach an $\varepsilon$-global minimum, we need at most

$$T \quad = \quad \mathcal{O}\left(\sqrt{\frac{LD^3}{\varepsilon}}\right),$$

iterations of the method, with the batches of size $b_g = \mathcal{O}(\frac{\sigma_g^2 D^2}{\varepsilon^2})$ and $b_h = \mathcal{O}(\frac{\sigma_h^2 D}{L\varepsilon})$ for the gradients and Hessians, respectively. Therefore, the total number of arithmetic operations needed to find an $\varepsilon$-global minimum is

$$\mathcal{O}\left(\frac{\sigma_g^2 L^{1/2} D^{3/2}}{\varepsilon^{5/2}} + d\frac{\sigma_h^2 D^{5/2}}{L^{1/2}\varepsilon^{3/2}}\right) \times GradCost.$$

For $\mu$-**uniformly convex functions of degree s = 3** (the case $\alpha = \frac{3}{2}$ and $\tau \sim \frac{1}{\sqrt{\mu}}$), using stochastic estimates (38) and setting $M = L$, we reach an $\varepsilon$-global minimum for any $\varepsilon > 0$ in at most

$$T \quad = \quad \mathcal{O}\left(\sqrt{\frac{L}{\mu}}\log(\frac{F_0}{\varepsilon})\right)$$

iterations of the method with the batches of sizes $b_g = \mathcal{O}(\frac{\sigma_g^2}{\mu^{2/3}\varepsilon^{4/3}})$ and $b_h = \mathcal{O}(\frac{\sigma_h^2}{\mu^{1/3}L\varepsilon^{2/3}})$. Therefore, the total number of arithmetic operations is bounded as

$$\tilde{\mathcal{O}}\left(\frac{\sigma_g^2\sqrt{L}}{\mu^{7/6}\varepsilon^{4/3}} + d\frac{\sigma_h^2}{\mu^{5/6}\sqrt{L}\varepsilon^{2/3}}\right) \times GradCost.$$

**$\mu$-strongly convex functions** ($\alpha = 2$ and $\tau \sim \frac{1}{\mu}$). For this class of functions, setting $M = L$, for any $\varepsilon > 0$ to get $\mathbb{E}[f(\boldsymbol{x}_{out})] - f^{\star} < \varepsilon$ we need to perform at most

$$T = \mathcal{O}\Big(t_0 + \log\log(\tfrac{\mu^3}{L^2\varepsilon})\Big),$$

with batches of size $b_g = \mathcal{O}(\frac{\sigma_g^2}{\mu\varepsilon})$ and $b_h = \mathcal{O}(\frac{\sigma_h^2}{L\sqrt{\mu\varepsilon}})$ and, where

$$t_0 = \tilde{\mathcal{O}}(\frac{\sqrt{L}F_0^{1/4}}{\mu^{3/4}}).$$

Therefore, the total number of arithmetic operations needed to find an $\varepsilon$-global minimum, in this case, is

$$\tilde{\mathcal{O}}\Big(\frac{t_0\sigma_g^2}{\mu\varepsilon} + d\frac{t_0\sigma_h^2}{L\sqrt{\mu\varepsilon}}\Big) \times GradCost.$$

### F.3 A special case of Theorem 4.3

Let us consider the special case $\alpha = 1$, which corresponds to minimizing *convex functions*. Theorem 4.3 implies the following global convergence rate:

$$\mathbb{E}[f(\boldsymbol{x}_{out})] - f^{\star} = \mathcal{O}\bigg(\frac{\delta_1 D^3}{S^2} + \frac{\delta_2 D^3}{S^2 m} + \frac{L D^3}{S^2 m^2}\bigg)$$

. We have the following special cases based on the choice of the helper functions.

- **Basic Cubic Newton** (Nesterov & Polyak, 2006) corresponds to $\delta_1 = \delta_2 = 0$. Thus, we get its known rate in the convex case. We reach an $\varepsilon$-global minimum using the following number of arithmetic operations:

$$\mathcal{O}\Big(\frac{nd}{\sqrt{\varepsilon}}\Big) \times GradCost. \tag{41}$$

- **Cubic Newton with Lazy Hessian updates** (Doikov et al., 2022) corresponds to $\delta_1 = 0, \delta_2 = L$. It gives $f(\boldsymbol{x}_{out}) - f^{\star} = \mathcal{O}\Big(\frac{LD^3}{S^2 m}\Big)$. By choosing $m = d$, we reach an $\varepsilon$-global minimum using the following number of arithmetic operations:

$$\mathcal{O}\Big(\frac{n\sqrt{d}}{\sqrt{\varepsilon}}\Big) \times GradCost. \tag{42}$$

  To the best of our knowledge, this complexity estimate of the Lazy Cubic Newton for convex functions is new, and it improves the complexity of the basic Cubic Newton (41) by factor $\sqrt{d}$.

- **Stochasic Cubic Newton with Variance Reduction** (Zhou et al., 2019; Wang et al., 2019) corresponds to sampling random batches $\mathcal{B}_g, \mathcal{B}_h$ of sizes $b_g, b_h$ respectively at each iteration, and setting $h_1 = \frac{1}{b_g}\sum_{i\in\mathcal{B}_g} f_i, h_2 = \frac{1}{b_h}\sum_{i\in\mathcal{B}_g} f_i$ in our helper framework. According to Lemma 3.7 we have $\delta_1 = \frac{L}{\sqrt{b_g}}$ and $\delta_2 = \tilde{\mathcal{O}}(\frac{L}{\sqrt{b_h}})$. Therefore, for $b_g \sim m^4, b_h \sim m^2$ and $M = L$, we get $\mathbb{E}[f(\boldsymbol{x}_{out})] - f^{\star} = \mathcal{O}(\frac{LD^3}{S^2 m^2})$. By choosing $m = (nd)^{1/5}1_{d\leq n^{2/3}} + n^{1/3}1_{d\geq n^{2/3}}$, we reach an $\varepsilon$-global minimum using the following number of arithmetic operations:

$$\mathcal{O}\Big(\frac{\min\{(nd)^{4/5}, n^{2/3}d+n\}}{\sqrt{\varepsilon}}\Big) \times GradCost. \tag{43}$$

  In Masiha et al. (2022), the global complexity estimate for the stochastic Cubic Newton with Variance Reduction was established for the gradient dominated functions of degree $\alpha = 1$. However, their result is different from ours, assuming only stochastic samples of the gradients and Hessians, while in our work, we allow recomputing the full gradient and Hessian once per $m$ iterations, similar in spirit to (Zhou et al., 2019; Wang et al., 2019). In our analysis, we take into account the total arithmetic cost of the operations. Hence, to the best of our knowledge, our complexity estimate (43) is new.

- **Stochastic Cubic Newton with Variance Reduction and Lazy Hessian updates.** In this case, by using sampling, we have $\delta_1 = \frac{L}{\sqrt{b_g}}$ and $\delta_2 = L$. If we take $m = (nd)^{1/3}1_{d \leq \sqrt{n}} + d1_{d \geq \sqrt{n}}$ then we reach an $\varepsilon$-global minimum using the following number of arithmetic operations:

$$\mathcal{O}\Big(\frac{\min\{(nd)^{5/6}, n\sqrt{d}\}}{\sqrt{\varepsilon}}\Big) \times GradCost.$$

This further improves the total complexity of both the method with Variance Reduction (43) and the Lazy Cubic Newton (42).

### F.4  Proof of Theorem 4.2

From Theorem C.1, we have

$$\mathbb{E}f(\boldsymbol{x}_t) - \mathbb{E}f(\boldsymbol{x}_{t+1}) \quad \geq \quad \frac{1}{1008\sqrt{M}}\mathbb{E}\|\nabla f(\boldsymbol{x}_{t+1})\|^{3/2} - \frac{4}{\sqrt{M}}\sigma_g^{3/2} - \frac{73}{M^2}\sigma_h^3.$$

By the definition of $(\tau, \alpha)$-gradient dominated functions we have

$$f(\boldsymbol{x}_t) - f^\star \quad \leq \quad \tau\|\nabla f(\boldsymbol{x}_t)\|^\alpha,$$

which leads to

$$\mathbb{E}\|\nabla f(\boldsymbol{x}_{t+1})\|^{3/2} \quad \geq \quad \mathbb{E}\Big(\tfrac{f(\boldsymbol{x}_{t+1})-f^\star}{\tau}\Big)^{\frac{3}{2\alpha}}.$$

If $\alpha \leq 3/2$, then by Jensen's inequality we have

$$\mathbb{E}\|\nabla f(\boldsymbol{x}_{t+1})\|^{3/2} \quad \geq \quad \Big(\tfrac{\mathbb{E}f(\boldsymbol{x}_{t+1})-f^\star}{\tau}\Big)^{\frac{3}{2\alpha}}. \tag{44}$$

For $\alpha > 3/2$ we need to assume that $\mathbb{E}f(\boldsymbol{x}_t) - f^\star \leq \tau\mathbb{E}[\|\nabla f(\boldsymbol{x}_t)\|]^\alpha$. This gives us (44).

We consider the sequence $F_t = \mathbb{E}f(\boldsymbol{x}_t) - f^\star$ and denote $\gamma = \frac{3}{2\alpha}$, $C = \frac{1}{1008\sqrt{M}\tau^\gamma}$ and $a = \frac{4}{\sqrt{M}}\sigma_g^{3/2} + \frac{25}{M^2}\sigma_h^3$. Then sequence $(F_t)_{t\geq 0}$ satisfies:

$$F_t - F_{t+1} \quad \geq \quad CF_{t+1}^\gamma - a. \tag{45}$$

We assume that $CF_{t+1}^\gamma - a \geq 0$ i.e., $F_{t+1} \geq \big(\frac{a}{C}\big)^{1/\gamma}$; we will prove that when this is the case, the sequence $(F_t)$ converges to $\big(\frac{a}{C}\big)^{1/\gamma}$; otherwise, we stop.

– **Case $\gamma = 1$.** Then
$$F_{t+1} \quad \leq \quad \tfrac{F_t+a}{C+1}.$$

From the recurrence, we have:

$$F_t \quad \leq \quad (1+C)^{-t}F_0 + \sum_{i=0}^{t-1}(1+C)^{-i}\tfrac{a}{1+C} \quad \leq \quad (1+C)^{-t}F_0 + \tfrac{a}{C}.$$

We note that $(1+C)^{-t} \leq \exp(\frac{-Ct}{1+C})$. Therefore,

$$F_t \quad \leq \quad \exp\Big(\tfrac{-Ct}{1+C}\Big)F_0 + \tfrac{a}{C}.$$

– **Case $1 < \gamma \leq 3/2$.** Then let $\tilde{F}_t = \frac{F_t}{C^{1/(1-\gamma)}}$ and $\tilde{a} = \frac{a}{C^{1/(1-\gamma)}}$ for which we have

$$\tilde{F}_t - \tilde{F}_{t+1} \quad \geq \quad \tilde{F}_{t+1}^\gamma - \tilde{a}.$$

Now let $x = \tilde{a}^{1/\gamma}$, and $\delta_t = \tilde{F}_t - x$. Then

$$\delta_t - \delta_{t+1} \quad \geq \quad (\delta_{t+1} + x)^\gamma - x^\gamma \quad \geq \quad \delta_{t+1}^\gamma,$$

where we used in the last inequality the fact that $(x + y)^\gamma \geq x^\gamma + y^\gamma$ for $\gamma \geq 1$ and $x, y \geq 0$. Here, we assume that $\delta_t \geq 0$. Otherwise, we have $F_t \leq (\frac{a}{C})^{1/\gamma}$.

Therefore, $\delta_t = \frac{F_t - (\frac{a}{C})^{1/\gamma}}{C^{1/(1-\gamma)}}$ and

$$\delta_t - \delta_{t+1} \geq \delta_{t+1}^\gamma.$$

We have

$$\frac{1}{(\gamma-1)\delta_{t+1}^{\gamma-1}} - \frac{1}{(\gamma-1)\delta_t^{\gamma-1}} = \frac{\delta_t^{\gamma-1} - \delta_{t+1}^{\gamma-1}}{(\gamma-1)\delta_t^{\gamma-1}\delta_{t+1}^{\gamma-1}}.$$

The concavity of $x \mapsto x^{\gamma-1}$ (since $\gamma \leq 2$) implies that for all $x, y \geq 0$ we have $x^{\gamma-1} - y^{\gamma-1} \geq (\gamma-1)x^{\gamma-2}(x-y)$. Hence,

$$\frac{1}{(\gamma-1)\delta_{t+1}^{\gamma-1}} - \frac{1}{(\gamma-1)\delta_t^{\gamma-1}} \geq \frac{\delta_t - \delta_{t+1}}{\delta_t \delta_{t+1}^{\gamma-1}} \geq \frac{\delta_{t+1}}{\delta_t}.$$

If $\delta_{t+1} \geq \delta_t/2$ then

$$\frac{1}{(\gamma-1)\delta_{t+1}^{\gamma-1}} - \frac{1}{(\gamma-1)\delta_t^{\gamma-1}} \geq 1/2,$$

otherwise, $\delta_{t+1} \leq \delta_t/2$, and in this case we have

$$\frac{1}{(\gamma-1)\delta_{t+1}^{\gamma-1}} - \frac{1}{(\gamma-1)\delta_t^{\gamma-1}} \geq \frac{1}{(\gamma-1)\delta_t^{\gamma-1}}(2^{\gamma-1} - 1) \geq \frac{2^{\gamma-1}-1}{(\gamma-1)\delta_0^{\gamma-1}},$$

where we used that $(\delta_t)_{t\geq 0}$ is decreasing. Thus, in all cases we have:

$$\frac{1}{(\gamma-1)\delta_{t+1}^{\gamma-1}} - \frac{1}{(\gamma-1)\delta_t^{\gamma-1}} \geq \min(1/2, \frac{2^{\gamma-1}-1}{(\gamma-1)\delta_0^{\gamma-1}}) := D.$$

By summing from $t = 0$ to $t = T - 1$, we get:

$$\frac{1}{(\gamma-1)\delta_T^{\gamma-1}} \geq DT.$$

In other words

$$\delta_T \leq \left(\frac{1}{(\gamma-1)DT}\right)^{\frac{1}{\gamma-1}}.$$

– **Case** $3/4 < \gamma < 1$. Then we have

$$F_{t+1} \leq \left(\frac{F_t - F_{t+1} + a}{C}\right)^{1/\gamma}.$$

By convexity of $x \mapsto x^{1/\gamma}$ we get

$$F_{t+1} \leq 2^{1/\gamma-1}\left(\frac{F_t - F_{t+1}}{C}\right)^{1/\gamma} + 2^{1/\gamma-1}\left(\frac{a}{C}\right)^{1/\gamma}.$$

Let $\delta_t = \frac{F_t - 2^{1/\gamma-1}(\frac{a}{C})^{1/\gamma}}{2^{1/\gamma}C^{1/(1-\gamma)}}$. Then we have

$$\delta_{t+1} \leq (\delta_t - \delta_{t+1})^{1/\gamma}.$$

The sequence $(\delta_t)_{t\geq 0}$ is decreasing, thus:

$$\delta_{t+1} \leq \delta_t^{1/\gamma}.$$

This is a superlinear rate, starting from the moment $\delta_t < 1$. We can show that from the very beginning $(\delta_t)_{t\geq 0}$ will decrease at least at a linear rate, and thus at some point we reach the region of superlinear convergence.

Indeed, we have $\frac{\delta_t}{\delta_{t+1}} \geq 1 + \frac{\delta_t - \delta_{t+1}}{\delta_{t+1}} \geq 1 + \frac{1}{\delta_{t+1}^{1-\gamma}} \geq 1 + \frac{1}{\delta_0^{1-\gamma}}$. Therefore,

$$\delta_{t+1} \leq (1 + \frac{1}{\delta_0^{1-\gamma}})^{-1}\delta_t = (1 - \frac{1}{1+\delta_0^{1-\gamma}})\delta_t \leq \exp(-\frac{1}{1+\delta_0^{1-\gamma}})\delta_t.$$

We reach $\delta_t \leq 1/2$ after $t \geq t_0 = (1 + \delta_0^{1-\gamma})\log(2\delta_0)$ iterations. After that $(t \geq t_0)$, we enjoy a superlinear convergence rate: $\delta_t \leq \left(\frac{1}{2}\right)^{\left(\frac{1}{\gamma}\right)^{t-t_0}}$. This finishes the proof. $\qquad\square$

### F.5 Proof of Theorem 4.3

In Theorem 4.3 we made the choice of updating the snapshot in the following way $\tilde{\boldsymbol{x}}_{s+1} = \boldsymbol{x}_{\arg\min_{i\in\{0,\cdots,m-1\}} f(\boldsymbol{x}_{sm+i})}$ which means that $f(\tilde{\boldsymbol{x}}_{s+1}) \leq f(\boldsymbol{x}_{sm+i})$ for all $i \in \{0,\cdots,m-1\}$.

For $s \in \{0,\cdots,S-1\}$ and $i \in \{0,\cdots,m-1\}$ We have the following inequality

$$\mathbb{E}f(\boldsymbol{x}_{sm+i}) - \mathbb{E}f(\boldsymbol{x}_{sm+i+1}) \geq \frac{1}{216\sqrt{M}}\mathbb{E}\|\nabla f(\boldsymbol{x}_{sm+i+1})\|^{3/2}$$
$$+ \mathbb{E}\Big[\frac{M}{72}r^3_{sm+i+1} - \big(\frac{4\delta_1^{3/2}}{\sqrt{M}} + \frac{73\delta_2^3}{M^2}\big)\|\boldsymbol{x}_{sm+i} - \boldsymbol{x}_{sm}\|^3\Big].$$

By the definition of gradient dominated functions we have $\mathbb{E}\|\nabla f(\boldsymbol{x}_{t+1})\|^{3/2} \geq \big(\frac{\mathbb{E}f(\boldsymbol{x}_{t+1})-f^\star}{\tau}\big)^{\frac{3}{2\alpha}}$.

So

$$\mathbb{E}f(\boldsymbol{x}_{sm+i}) - \mathbb{E}f(\boldsymbol{x}_{sm+i+1}) \geq \frac{1}{216\sqrt{M}}\big(\frac{\mathbb{E}f(\boldsymbol{x}_{sm+i+1})-f^\star}{\tau}\big)^{\frac{3}{2\alpha}}$$
$$+ \mathbb{E}\Big[\frac{M}{72}r^3_{sm+i+1} - \big(\frac{4\delta_1^{3/2}}{\sqrt{M}} + \frac{73\delta_2^3}{M^2}\big)\|\boldsymbol{x}_{sm+i} - \boldsymbol{x}_{sm}\|^3\Big]$$
$$\geq \frac{1}{216\sqrt{M}}\big(\frac{\mathbb{E}f(\tilde{\boldsymbol{x}}_{s+1})-f^\star}{\tau}\big)^{\frac{3}{2\alpha}}$$
$$+ \mathbb{E}\Big[\frac{M}{72}r^3_{sm+i+1} - \big(\frac{4\delta_1^{3/2}}{\sqrt{M}} + \frac{73\delta_2^3}{M^2}\big)\|\boldsymbol{x}_{sm+i} - \boldsymbol{x}_{sm}\|^3\Big].$$

Summing the above inequality for $0 \leq i \leq m-1$ and remarking that $\tilde{\boldsymbol{x}}_s = \boldsymbol{x}_{sm}$, we get

$$\mathbb{E}f(\tilde{\boldsymbol{x}}_s) - \mathbb{E}f(\boldsymbol{x}_{(s+1)m}) \geq \frac{m}{216\sqrt{M}}\big(\frac{\mathbb{E}f(\tilde{\boldsymbol{x}}_{s+1})-f^\star}{\tau}\big)^{\frac{3}{2\alpha}}$$
$$+ \mathbb{E}\Big[\sum_{i=0}^{m-1}\frac{M}{72}r^3_{sm+i+1} - \big(\frac{4\delta_1^{3/2}}{\sqrt{M}} + \frac{73\delta_2^3}{M^2}\big)\|\boldsymbol{x}_{sm+i} - \boldsymbol{x}_{sm}\|^3\Big].$$

By the definition of $\tilde{\boldsymbol{x}}_{s+1}$ we have $f(\tilde{\boldsymbol{x}}_{s+1}) \leq f(\tilde{\boldsymbol{x}}_{sm+i})$ for all $i \in \{0,\ldots,m-1\}$ which leads to

$$\mathbb{E}f(\tilde{\boldsymbol{x}}_s) - \mathbb{E}f(\tilde{\boldsymbol{x}}_{s+1}) \geq \frac{m}{216\sqrt{M}}\big(\frac{\mathbb{E}f(\tilde{\boldsymbol{x}}_{s+1})-f^\star}{\tau}\big)^{\frac{3}{2\alpha}}$$
$$+ \mathbb{E}\Big[\sum_{i=0}^{m-1}\frac{M}{72}r^3_{sm+i+1} - \big(\frac{4\delta_1^{3/2}}{\sqrt{M}} + \frac{73\delta_2^3}{M^2}\big)\|\boldsymbol{x}_{sm+i} - \boldsymbol{x}_{sm}\|^3\Big].$$

For $M$ satisfying (39) we have $\sum_{i=0}^{m-1}\frac{M}{72}r^3_{sm+i+1} - \big(\frac{4\delta_1^{3/2}}{\sqrt{M}} + \frac{73\delta_2^3}{M^2}\big)\|\boldsymbol{x}_{sm+i} - \boldsymbol{x}_{sm}\|^3 \geq 0$ thus we have

$$\mathbb{E}f(\tilde{\boldsymbol{x}}_s) - \mathbb{E}f(\tilde{\boldsymbol{x}}_{s+1}) \geq \frac{m}{216\sqrt{M}}\big(\frac{\mathbb{E}f(\tilde{\boldsymbol{x}}_{s+1})-f^\star}{\tau}\big)^{\frac{3}{2\alpha}}.$$

Let us define $F_s = \mathbb{E}f(\tilde{\boldsymbol{x}}_s) - f^\star$. Then

$$F_s - F_{s+1} \geq CF_{s+1}^\gamma,$$

which is a special case of inequality (45) with $a = 0$. Thus we can apply our findings from before and replace $a$ with 0. $\qquad\square$

