# OpenReview forum: "Unified Convergence Theory of Stochastic and Variance-Reduced Cubic Newton Methods"
_TMLR — Accepted by TMLR_

### Review · Reviewer_jpd1 · 2024-04-14

**Summary Of Contributions:**

This paper presents a helper framework for stochastic Cubic Newton methods in optimization, specifically addressing non-convex minimization problems. The primary contributions include:

1) This paper introduces a new framework for stochastic and variance-reduced second-order algorithms. This framework allows for flexibility in batch sizes and incorporates noisy, potentially biased estimates of gradients and Hessians.

2) It provides global complexity guarantees and recovers the best-known complexities for stochastic and variance-reduced Cubic Newton methods under weak assumptions on the noise.

3) The authors develop new algorithms using lazy stochastic second-order methods that improve arithmetic complexity, particularly in large dimension scenarios.

4) The authors also demonstrate that auxiliary tasks can improve learning outcomes if a particular similarity measure is small.

**Audience:**

Yes

**Claims And Evidence:**

Yes

**Requested Changes:**

1) Additional empirical results are expected, especially on more complicated tasks, which would strengthen the claims about the framework's effectiveness and practicality.

2) More discussion about how to design or find a good helper. Also, it's better to give more discussion about the assumption used, as the current assumptions seem somewhat strong.

**Strengths And Weaknesses:**

Strengths:

1) This paper offers rigorous theoretical guarantees with clear complexity bounds and improvements over existing methods.

2) This paper is clearly written and well organized. The authors discuss the related work properly.

3) The helper framework is a novel approach that may influence future research in stochastic optimization strategies.

Weaknesses:

1) The authors mainly validate their methods via logistic regression problems. More results on more complicated tasks and datasets are expected, since the theoretical improvements need more empirical results to validate their effectiveness.

2) Although a helper framework can enhance performance, how to devise a good helper may be a problem in practice. It would be better if the authors could discuss more about this.

3) The effectiveness of the approach heavily relies on the assumptions about noise and data similarity, which might not always hold in practical scenarios.

---

> ### Author Response · Authors · 2024-05-25
>
> Dear Reviewer,
>
> Thank you very much for your time and expertise invested in reviewing our work, and for the positive feedback. We hope that we addressed all raised issues in our rebuttal.
>
> 1-**Additional experiments.**
>
> We provide additional results comparing our second-order methods with standard Gradient Descent (GD), Stochastic Gradient Descent (SGD), and the Adam method for minimizing the Diagonal Neural Network model, which serves as a good theoretical model for studying the non-convex landscape of deep learning (Woodworth et al., 2020; Pesme & Flammarion, 2023) – please, see Figure 5. We observe that the stochastic first-order methods, including Adam, struggle with slow convergence, whereas our second-order algorithms achieve higher accuracy with improved arithmetic cost and total computational time. The most advanced version of our method (Lazy VR) achieves the best performance on this task.
>
> We also include a comparison with the Adam method in our experiments with Logistic Regression with both convex and non-convex regularizers on several practical datasets – please, see Figures 1,3,4.
>
> Additionally, we are adding new experiments for our methods, solving Matrix Factorization problems. We are open to conducting any additional experiments, based on your suggestions. Thanks.
>
> We would like to emphasize that our work is primarily theoretical, unifying known state-of-the-art stochastic second-order methods for non-convex problems into a general helper framework, and proposing new schemes that achieve the best-known convergence rates and the total arithmetic complexity (Table 1).
>
> 2- **Examples of helpers and assumptions.**
>
> The main examples of our helper framework are the following ones:
>
> - The basic helpers (9) along with simple formulas for gradient and Hessian estimation (6). This approach is analyzed under Assumption 3.1. We include additional remarks after the assumption to highlight that it is related to standard assumptions from the literature on stochastic optimization (bounded variance and Lipschitzness of the gradient).
>
> - Advanced gradient and Hessian estimation (11), (12) that recovers the variance reduction and can be used in several situations. We prove general convergence for this estimation under Assumption 3.4. We have a simple sufficient condition for this assumption to hold, which is Lipschitzness of the Hessian for the helpers (this is briefly discussed after Assumption 3.4). This estimation includes the following examples:
>    - General variance reduction (16), which recovers the known stochastic second-order methods from the literature.
>
>    - Variance reduction with lazy Hessians (18), which allows to improve the arithmetic complexity of the general variance reduction (16).
>
> 	These are the most important examples from our work. For both of them Assumption 3.4 	is satisfied, if the components f_i of our objective have Lipschitz continuous Hessian.
>
>    - Additionally, we include an explicit expression for the helper in core sets application (20). We also provide Example 3.8, demonstrating when Assumption 3.4 is satisfied.
>
>    - Finally, we add an explicit formula for the helper in applications with semi-supervised learning (22). We also demonstrate that Assumption 3.4 is satisfied for the linear models with logistic loss (Example 3.9).

---

### Review · Reviewer_eFK5 · 2024-04-30

**Summary Of Contributions:**

Summary of the main arguments:

This paper presents a unified framework called the "helper framework" that provides a unified view of stochastic and variance-reduced second-order algorithms equipped with global complexity guarantees. The helper framework offers flexibility for constructing and analyzing stochastic Cubic Newton methods, allowing arbitrary batch sizes and the use of noisy and possibly biased estimates of gradients and Hessians.

The authors claim that the paper recovers the best-known complexities for stochastic and variance-reduced Cubic Newton methods under weak assumptions on the noise.

It also introduces a new lazy stochastic second-order method that significantly improves the arithmetic complexity for large-dimensional problems. The paper establishes complexity bounds for gradient-dominated objectives, including convex and strongly convex problems. For auxiliary learning, the authors show that using a helper function can outperform training alone if a given similarity measure is small.

Assumptions used by the authors:

1\. The Hessian of the objective function f is Lipschitz continuous (Assumption 1).

2\. There is bounded similarity between the gradient and Hessian estimates and the true values (Assumption 2).

3\. There is Lipschitz similarity between the gradient and Hessian estimates and the true values (Assumption 3).

4\. The objective function exhibits gradient dominance (Assumption 4).

Contributions of the paper:

1\. Introducing the helper framework that unifies multiple methods, including stochastic methods, variance reduction, lazy methods, core sets, and semi-supervised learning.

2\. Recovering previous versions of the variance-reduced stochastic Cubic Newton methods with known rates and providing new algorithms that employ lazy Hessian updates, improving the arithmetic complexity for high-dimensional problems.

3\. Proving a benefit from using auxiliary tasks as helpers in the framework, potentially replacing the smoothness constant with a smaller similarity constant.

4\. Analyzing both general non-convex functions and gradient-dominated functions, including convex and uniformly convex functions.

5\. Justifying new improved complexity bounds for the deterministic Cubic Newton method with lazy Hessian updates and for the stochastic Cubic Newton algorithms with variance reduction, taking into account the total arithmetic cost of the operations.

**Audience:**

Yes

**Broader Impact Concerns:**

This work is primarily theoretical in nature. Hence, there is no identifiable potential for negative societal impact arising from this work.

**Claims And Evidence:**

No

**Requested Changes:**

1) Could you please specify the difference between this work and several other recent advancements in this field?
(1) Hanzely, Slavomír. "Sketch-and-Project Meets Newton Method: Global $\mathcal O (k^{-2}) $ Convergence with Low-Rank Updates." arXiv preprint arXiv:2305.13082 (2023).
(2) Mishchenko, Konstantin. "Regularized Newton Method with Global Convergence." SIAM Journal on Optimization 33.3 (2023): 1440-1462.
(3) Hanzely, Slavomír, et al. "A Damped Newton Method Achieves Global $\mathcal O\left (\frac {1}{k^ 2}\right) $ and Local Quadratic Convergence Rate." Advances in Neural Information Processing Systems 35 (2022): 25320-25334.

2) "while it can be also possible to generalize our results to functions represented as expectation over an arbitrary probability distribution."
I think it is better to remove this phrase as these results are not presented in the work.

3)Considering the experiments, could the authors please comment on why in Figure 1 and Figure 3 the stochastic cubic Newton method has almost zero arithmetic computations?

**Strengths And Weaknesses:**

Strengths:
1) The paper provides a unified framework that encompasses multiple methods, offering a comprehensive view of stochastic and variance-reduced second-order algorithms.
2) The paper introduces new algorithms, such as the lazy stochastic second-order method, that improve upon existing methods in terms of arithmetic complexity.
3) The paper establishes complexity bounds for both general non-convex functions and gradient-dominated functions, providing a thorough analysis of the proposed methods.
4) Overall, the text is well-written, and the main points are well-expressed.

Weaknesses:
1) It should be noted that there are several closely related papers in the area of cubic Newton methods (even in a convex regime), which unfortunately are overlooked. See "Requested Change 1)".
2) In some places, the paper lacks clarity. For example, at the end of the fourth page, the authors introduce a general principle involving a computationally cheap function h and an expensive function f - h. Then they suddenly mention the need for two helpers h\_1 and h\_2. This leads to a slight loss of intuition and a certain misalignment with what is written in the general principle. Could the authors please clarify this point?
3) According to the text, the "unified property" of the analysis is given through auxiliary helper functions h\_1 and h\_2. However, in section 3.4, the discussion of these functions is mostly hand-waving and lacks a rigorous definition of h\_1 and h\_2 (for each particular example), as well as a rigorous proof stating the conditions under which delta\_1 and delta\_2 satisfy the assumptions. However, it is important to point out that these assumptions are satisfied for the subsampling of functions f\_i. Nonetheless, similarly to what I mentioned earlier, it is unclear whether this framework is useful outside this special case.
4) Regarding real-time comparison, I should note that this comparison in real-time is hardware implementation dependent. Without reviewing the source code, which is not disclosed, it is difficult to determine how well the algorithms are implemented.

---

> ### Author Response · Authors · 2024-05-25
>
> Dear Reviewer,
>
> Thank you very much for your time and expertise invested in reviewing our work, and for the positive feedback. We hope that we addressed all raised issues in our rebuttal.
>
> 1. **Additional references.**
>
> We appreciate the additional references, which we include in the new version of our paper. The main difference between our work and the referenced studies is that we focus on studying the Cubically regularized version of the Newton method. This approach allows us to establish global complexity bounds for general classes of problems, including those with non-convex objectives (Section 3). In contrast, the main results from (Hanzely, 2023), (Mishchenko, 2023), and (Hanzely et al, 2022) assume that the objective is convex, allowing the use of different regularization techniques, such as quadratic regularization or damped Newton steps. Additionally, our work considers the setting of a large dataset size, where computing the full gradient and the Hessian at every iteration is too expensive. Consequently, our framework employs stochastic oracle information in the second-order model. We add this discussion, along with the new references, to the revised version of the manuscript. Thanks.
>
> 2- **The need for two helpers h_1 and h_2 instead of one.**
>
> The use of two different helper functions provides us with extra flexibility in the design of the method, while it is still possible to consider for simplicity only one helper, i.e. h_1 = h_2. However, separating the helpers allows us to take into account the difference between arithmetical cost for the first-order and second-order information. We add this additional explanation to the paper.
>
>
> 3- **Examples of helpers and assumptions.**
>
>
> The main examples of our helper framework are the following ones:
>
> - The basic helpers (9) along with simple formulas for gradient and Hessian estimation (6). This approach is analyzed under Assumption 3.1. We include additional remarks after the assumption to highlight that it is related to standard assumptions from the literature on stochastic optimization (bounded variance and Lipschitzness of the gradient).
>
> - Advanced gradient and Hessian estimation (11), (12) that recovers the variance reduction and can be used in several situations. We prove general convergence for this estimation under Assumption 3.4. We have a simple sufficient condition for this assumption to hold, which is Lipschitzness of the Hessian for the helpers (this is briefly discussed after Assumption 3.4). This estimation includes the following examples:
>    - General variance reduction (16), which recovers the known stochastic second-order methods from the literature.
>
>    - Variance reduction with lazy Hessians (18), which allows to improve the arithmetic complexity of the general variance reduction (16).
>
>
>    These are the most important examples from our work. For both of them Assumption 3.4 	is satisfied, if the components f_i of our objective have Lipschitz continuous Hessian.
>    - Additionally, we include an explicit expression for the helper in core sets application (20). We also provide Example 3.8, demonstrating when Assumption 3.4 is satisfied.
>
>    - Finally, we add an explicit formula for the helper in applications with semi-supervised learning (22). We also demonstrate that Assumption 3.4 is satisfied for the linear models with logistic loss (Example 3.9).
>
>
> 4- **Real-time profiling.**
> In order to distinguish the effects of our specific hardware instance, we also include a new graph with detailed time profiling of our implementation of the oracle for the Diagonal Neural Network model – please, see Figure 6. We see that the average cost of computing the Hessian is significantly higher than the cost of computing a single gradient, and the difference grows approximately linearly with d (the dimension of the problem). This plot motivates our developments in stochastic and lazy Hessian computations, which are supported both theoretically and empirically.
>
>
> 5- **Figure 1 and Figure 3:** the stochastic cubic Newton had almost zero arithmetic computations due to improper scaling of the graph. We apologize for that and provide a new version of the plot (the same run) in log scale  – please, see the new Figures 1,3 and 4. In the right plot we demonstrate the total computational time (for our hardware implementation), while in the left plot, we show the convergence with respect to the total arithmetic computation of the oracles (gradients and Hessians). We observe that the stochastic first-order methods struggle with slow convergence, whereas our second-order algorithms achieve higher accuracy with improved arithmetic cost and total computational time. The most advanced version of our method (Lazy VR) achieves the best performance.
>
> 6-**Minor:** The phrase regarding "expectation over an arbitrary probability distribution" has been removed. Thanks.

---

### Review · Reviewer_Jadv · 2024-05-12

**Summary Of Contributions:**

This paper proposes a novel framework that unifies stochastic and variance-reduced second-order algorithms with global complexity guarantees, applying to both general non-convex problems and specific machine learning tasks. The introduction of the helper framework is innovative, potentially impacting the design and analysis of stochastic Cubic Newton methods.

**Audience:**

Yes

**Claims And Evidence:**

Yes

**Requested Changes:**

1. Experiments on more convincing datasets.
2. Comparasion with more advanced first-order methods.

**Strengths And Weaknesses:**

Strengths:

The theoretical development is sound and the mathematical formulations are rigorously presented.

Weakness:

1. The current experimental validation primarily uses datasets from the LIBSVM library, which are relatively simplistic and might not fully demonstrate the scalability and efficacy of the proposed methods in challenging scenarios.
2. The first-order baselines are too weak. Consider comparing with those empirically better ones, like Adam and AdamW. Also, I think it would be better to show the detailed profiling, like the time for computing the gradient and hessian respectively.

---

> ### Author Response · Authors · 2024-05-25
>
> Dear Reviewer,
>
> Thank you very much for your time and expertise invested in reviewing our work, and for the positive feedback. We hope that we addressed all raised issues in our rebuttal.
>
>  - **Additional experiments:**
>
> We provide additional results comparing our second-order methods with standard Gradient Descent (GD), Stochastic Gradient Descent (SGD) and the Adam method for minimizing the Diagonal Neural Network model, which serves as a good theoretical model for studying the non-convex landscape of deep learning (Woodworth et al., 2020; Pesme & Flammarion, 2023) – please, see Figure 5. We observe that the stochastic first-order methods, including Adam, struggle with slow convergence, whereas our second-order algorithms achieve higher accuracy with improved arithmetic cost and total computational time. The most advanced version of our method (Lazy VR) achieves the best performance on this task.
>
> We also include a comparison with the Adam method in our experiments with Logistic Regression with both convex and non-convex regularizers on several practical datasets – please, see Figures 1,3,4.
>
> Additionally, we are adding new experiments for our methods, solving Matrix Factorization problems. We are open to conducting any additional experiments, based on your suggestions. Thanks.
>
> We would like to emphasize that our work is primarily theoretical, unifying known state-of-the-art stochastic second-order methods for non-convex problems into a general helper framework, and proposing new schemes that achieve the best-known convergence rates and the total arithmetic complexity (Table 1).
>
>  - **Detailed profiling:**
>
> We include a new graph with detailed time profiling of our implementation of the oracle for the Diagonal Neural Network model – please, see Figure 6. We see that the average cost of computing the Hessian is significantly higher than the cost of computing one gradient, and the quotient grows approximately linearly with d (the dimension of the problem). This plot motivates our developments in stochastic and lazy Hessian computations, which are supported both theoretically and empirically.

---

### Author Response · Authors · 2024-05-30
**New experiments on MNIST and CIFAR10**

Dear reviewers,

We would like to inform you that we have included two additional experiments in the appendix (Figures 8 and 9) related to the MNIST and CIFAR10 datasets. Both experiments are logistic regression experiments, but they clearly show the advantages our method has, especially in terms of the time it saves and the better error it reaches. We note also that we are trying to add an experiment using a simple neural network.

---

### Author Response · Authors · 2024-06-15
**Additional experiment**

Dear Reviewers,
We want to inform you that we have added a new experiment (Figure 10 in the Appendix) using a simple two-layer NN.
The experiment is also favorable to our algorithm.

---

### Decision · Action_Editor_exy7 · 2024-07-30

**Recommendation:** Accept as is

**Comment:**

The authors effectively demonstrated the usefulness of the helper framework by recovering the best-known complexity of variance-reduced cubic Newton methods and proposing a new method. The introduction of this framework has the potential to impact the design and analysis of stochastic cubic Newton methods.

All concerns raised by reviewers have been well addressed. Considering the above contributions, this paper meets the acceptance criteria of TMLR.

**Audience:**

This paper could be of interest to parts of the TMLR audience.

**Claims And Evidence:**

This paper presents the "helper framework," which provides a unified view of stochastic and variance-reduced second-order algorithms with global complexity guarantees. This framework allows flexibility in batch sizes and incorporates noisy, potentially biased estimates of gradients and Hessians.

Moreover, using helper framework, the authors achieve the best-known complexities for stochastic and variance-reduced cubic Newton methods under weak noise assumptions and develop new algorithms that improve arithmetic complexity for large-dimensional problems.